# EF-VLA: VISION-LANGUAGE-ACTION MODELS WITH ALIGNED VISION LANGUAGE FEATURES FOR BETTER GENERALIZATION

## ABSTRACT

Recent advances in Vision-Language-Action (VLA) models can enable robots to perform a wide range of tasks based on language or goal-based instructions. These VLA models typically encode text and images into disjoint tokens, generating actions that align with the given instructions. This requires the VLA models to simultaneously perform vision-language understanding and precise closed-loop control, resulting in significant challenges for them to generalize to new environments. However, contrastive pre-trained VLMs, such as CLIP, already possess vision-language alignment capabilities, which are underutilized by current VLA models. In this paper, we propose Early Fusion VLA (EF-VLA), a novel VLA architecture that exploits CLIP's vision-language understanding by performing *early fusion*, extracting fine-grained vision-language tokens relevant to the task instructions before passing them to the transformer policy. EF-VLA keeps the VLM frozen, allowing it to effectively perform unseen tasks without requiring fine-tuning, which often reduces generalization capabilities. Simulation and real-world experiments suggest that EF-VLA outperforms state-of-the-art VLA models on diverse tasks, with significant generalization capabilities in unseen environments.

## 1 INTRODUCTION

Recent advancements in Large Language Models (LLMs) and Vision-Language Models (VLMs) have inspired the exploration of scaling datasets and computational resources for vision-language-action (VLA) models (Collaboration et al., 2024; Khazatsky et al., 2024; Octo Model Team et al., 2024; Kim et al., 2024). Different input modalities are usually encoded into separate tokens: multi-view images encoded via visual feature extractors, along with tokenized language instructions, optionally with the robot's proprioceptive states, are fed into a transformer-based robot policy for end-to-end action generalization. This approach requires the policy network to connect the vision and language information and conduct precise robot control, which often presents significant challenges, especially in unseen environments.

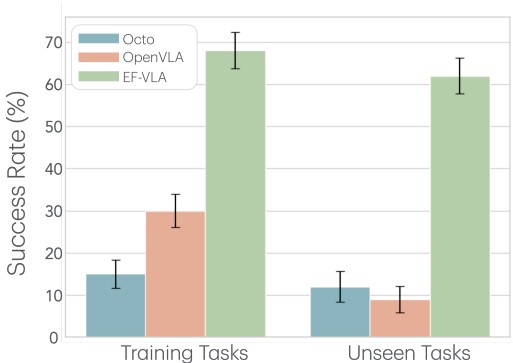

Figure 1: **Real-world Robot Experiments.** EF-VLA demonstrates significantly higher success rates on both training and unseen real-world tasks compared to Octo and OpenVLA. EF-VLA exhibits better generalization to unseen objects, maintaining strong performance across a variety of novel tasks. Error bars represent the standard error calculated over 100 runs across 10 training tasks and 70 runs across 7 unseen tasks.

Numerous works (Brohan et al., 2023; Kim et al., 2024) have demonstrated the benefits of using pre-trained vision encoders or vision-language models in robotics. While these approaches already use the rich visual features extracted from pre-trained vision encoders, the policy network—often a fine-tuned language model or a transformer trained from scratch—must still learn to associate the language instructions with the visual information. However, models like CLIP (Radford et al., 2021) and SigLIP (Zhai et al., 2023) are already trained to align image and text instructions, with

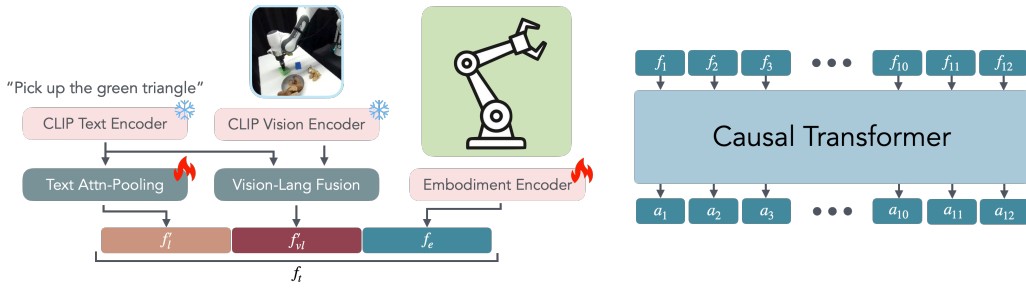

(a) Multi-Modal State Representation

(b) Transformer-Based Policy

**Figure 2: Model architecture of EF-VLA.** At each timestep $t$, vision and language features are extracted by a pre-trained CLIP model and fused into a set of tokens $f_{vl}$ (see Figure 3). The fused vision-language tokens $f_{vl}$ and the text tokens $f_l$ are each processed through separate attention pooling layers, producing two single tokens $f'_{vl}$ and $f'_l$, respectively. The robot's proprioception is encoded by an embodiment encoder to generate the embodiment representation $f_e$. The tokens $f'_l$, $f'_{vl}$, and $f_e$ are then concatenated along the channel dimension to form $f_t$, which serves as input to a causal transformer. Based on a context window of 12 steps, the model autoregressively predicts the next 12 actions ($a_t$) at each step.

an impressive performance on various downstream tasksIt can even perform more fine-grained tasks like open-vocabulary segmentation, by extracting fine-grained patch-level correspondence in recent works (Rao et al., 2022; Lan et al., 2024; Dong et al., 2023). Given the capabilities of these VLMs, it's redundant for the policy network to learn the vision-language alignment from scratch, particularly since robot datasets are far less semantically diverse compared to large vision-language datasets (Schuhmann et al., 2022) where these VLMs are trained on. Additionally, despite the effort these large VLAs to generalize to unseen tasks, there still exists a performance discrepancy between training tasks and unseen tasks. Some prior works such as OpenVLA (Kim et al., 2024) have shown that fine-tuning the vision encoder is critical for improving its performance on new tasks. However, fine-tuning, especially for language-aligned encoders like CLIP, introduces a critical trade-off: it can impair generalization and long-tail classification performance (Kerr et al., 2023; Rashid et al., 2023; Lan et al., 2024), posing notable over-fitting issues.

We seek to preserve the generalization capabilities of VLMs for effective performance under unseen scenarios. To this end, we propose Early Fusion VLA (EF-VLA), a novel VLA architecture that exploits VLM's vision-language understanding by performing *early fusion*. Specifically, we refer *early fusion* to the vision language alignment before the policy transformer, whereas *late fusion* refers to vision language alignment in a relatively later stage, in the policy transformer. While in principle, any VLMs with strong vision-language alignment capabilities can be applicable, in this paper, we utilize CLIP, due to its wide usage and strong vision-language alignment capability. Furthermore, recent work ClearCLIP (Lan et al., 2024) allows the extraction of fine-grained and semantically meaningful vision-language features, necessary for guiding the robot policy to generate accurate actions. We adopt the architecture from ClearCLIP, where we directly use the clean text-patch correspondence as our frozen vision-language representations, preserving the inherent vision-language understanding ability of the CLIP to a large extent.

Figure 2 provides an overview of EF-VLA. EF-VLA obtains the fused vision language features from ClearCLIP. The policy network receives the fused vision-language token, a language token, and the proprioception token to autoregressively predict actions in a causal transformer. Intuitively, the fused vision language features provide task related vision information such as object locations. The policy network then plans the action based on the provided object location, task information and the robot state. Importantly, we keep the CLIP model frozen during training to preserve its pre-trained powerful vision-language alignment. Both physical and simulation experiments show that EF-VLA significantly outperforms existing VLA models, demonstrating superior generalization to novel objects and environments with minimal performance degradation (Figure 1).

To summarize, our contributions are:

1. we propose EF-VLA, a VLA model that performs fine-grained early-fusion of vision and language information. It leverages a pre-trained CLIP model with ClearCLIP architecture to extract fine-grained vision-language features for effective performance on robotic tasks.

2. EF-VLA can outperform the state-of-the-art VLA models and its ablations on diverse robot manipulation tasks. More significantly, EF-VLA can perform unseen tasks in a zero-shot

manner without the need to finetune vision encoders, which maximally preserves and leverages the superior generalization capabilities of pre-trained vision-language models.

## 2 RELATED WORK

### 2.1 VISION LANGUAGE PRE-TRAINING

Vision-language pre-training (VLP) seeks to improve the performance of downstream tasks that involve both vision and language by training models on extensive datasets of image-text pairs. A prominent class of vision-language models leverages contrastive learning (Alayrac et al., 2020; Cherti et al., 2023; Jia et al., 2021; Radford et al., 2021; Yao et al., 2021; Yuan et al., 2021; Zhai et al., 2023). Among them, CLIP (Radford et al., 2021), which was trained on a private WIT-400M dataset of image-text pairs, demonstrates impressive zero-shot capabilities across various downstream tasks, including image-text retrieval and image classification through text prompts. Furthermore, CLIP shows potential for application in broader fields such as decision making and robotics, where robots are required to perform language-specified tasks based on visual inputs.

Recent early-fusion approaches, exemplified by BLIP (Li et al., 2022; 2023), extract visual features using a language-aligned vision model and apply multilayered cross-attention between encoded language features and visual features. The resulting features are then passed into a language model. However, many researchers have observed that fine-tuning or even applying additional layers on top of CLIP (instead of using raw CLIP features) (Kerr et al., 2023; Lan et al., 2024) may result in models with weaker reasoning capabilities compared to vanilla CLIP.

### 2.2 VISION LANGUAGE ACTION MODELS

In recent years, there has been a surge of interest in developing robot foundation models, largely inspired by the success of large language models (LLMs) and vision-language models (VLMs) (Devlin et al., 2018; Radford et al., 2018; 2019; Brown et al., 2020; Chowdhery et al., 2023; Achiam et al., 2023; Radford et al., 2021; Li et al., 2023). A key hypothesis driving this trend is that more capable robot foundation models can emerge by scaling up robot datasets, increasing model capacity, and co-training or pre-training models on vision and language datasets. This has led researchers in the robot learning community to train robot foundation models, investigate pre-training strategies, and iterate on model designs (Brohan et al., 2022; 2023; Kim et al., 2024; Octo Model Team et al., 2024; Jang et al., 2022; Jiang et al., 2023; Reed et al., 2022; Collaboration et al., 2024; Shah et al., 2023; Fu et al., 2024).

Many existing VLMs (Liu et al., 2023; Laurençon et al., 2024; Karamcheti et al., 2024) use a "late-fusion" approach, where visual features and languages are directly passed into the LLM to generate answers. Similarly, the majority of Vision-Language-Action (VLA) models also opt for late-fusion, where language, vision, and robot proprioception data are separately encoded by modality-specific feature extractors before being fed into a single transformer policy. This method has shown promise in many language-conditioned multi-task learning models (Jiang et al., 2023; Brohan et al., 2023; Jang et al., 2022; Reed et al., 2022; Collaboration et al., 2024; Shah et al., 2023), including current open-source state-of-the-art models such as Octo (Octo Model Team et al., 2024) and OpenVLA (Kim et al., 2024).

In contrast to the late-fusion approach, "early-fusion" combines vision and language inputs before feeding them into the language model or during visual feature extraction. Early works such as FiLM (Perez et al., 2018) encode text information and fuse these features into each block of a ResNet (He et al., 2016). RT-1 (Brohan et al., 2022), one of the first language-conditioned robot models, uses FiLM to encode text information for action generation. However, FiLM and RT-1 need to learn the language-vision alignment from task data, thus cannot leverage pre-trained models such as CLIP (Radford et al., 2021), where visual features are already aligned with text.

Inspired by ClearCLIP (Lan et al., 2024), EF-VLA distinguishes itself by using a similarity-based fusion between visual patch features and text token features from CLIP while also incorporating additional text tokens and robot embodiment tokens as inputs to the robot policy. This approach allows us to leverage the strengths of fine-grained features from the pre-trained vision-language models while maintaining the flexibility to incorporate robot-specific information.

## 3 METHOD

We propose Early Fusion VLA, a vision-language-action model for learning a robot manipulation policy through early fusion on the vision-language features. We first describe how EF-VLA employs early-fusion between the vision and language modalities, then provide a more detailed explanation of the model architecture.

### 3.1 VISION-LANGUAGE EARLY FUSION

EF-VLA utilizes a pre-trained CLIP for vision-language fusion. Consider a ViT-based CLIP vision encoder (Radford et al., 2021) consisting of a series of residual attention blocks. Each of these blocks takes as input a collection of visual tokens $X = [x_{\text{cls}}, x_1, \ldots, x_{h \times w}]^T$, where $x_{\text{cls}}$ represents the learnable global class token, and outputs the feature $X_{out}$ as shown below:

$$q = \text{Proj}_q(\text{LN}(X)), \quad k = \text{Proj}_k(\text{LN}(X)), \quad v = \text{Proj}_v(\text{LN}(X)) \tag{1}$$

$$X_{\text{sum}} = X + X_{\text{attn}} = X + \text{Proj}(\text{Attn}(q, k, v)) \tag{2}$$

$$X_{\text{out}} = X_{\text{sum}} + \text{FFN}(\text{LN}(X_{\text{sum}})) \tag{3}$$

Proj, LN, and FFN denote linear projection matrix, layer norm (Ba, 2016), and feed-forward network respectively. A recent work ClearCLIP (Lan et al., 2024) shows improved training-free open-vocabulary segmentation performance by using CLIP's last self-attention block's attention feature $X_{\text{attn}}$ instead of the CLIP's output feature $X_{\text{out}}$, resulting in segmentation with less noise. Inspired by ClearCLIP, we use a parameter-free method to extract task-relevant CLIP features.

In EF-VLA, we extract text per-token features from CLIP's language encoder $f_l$ ($m$ tokens). For the visual features, motivated by the improved ability of ClearCLIP to capture text-aligned visual features, we specifically utilize the attention output $X_{\text{attn}}$ from the last vision attention layer, rather than the CLIP's output feature $X_{\text{out}}$, denoting it as $f_v$ (n tokens), where $n = h \times w$ is the total number of patch tokens from ViT. Figure 6 demonstrates how using $X_{\text{attn}}$ enhances the alignment between visual features and language semantics, illustrating the effectiveness of this approach.

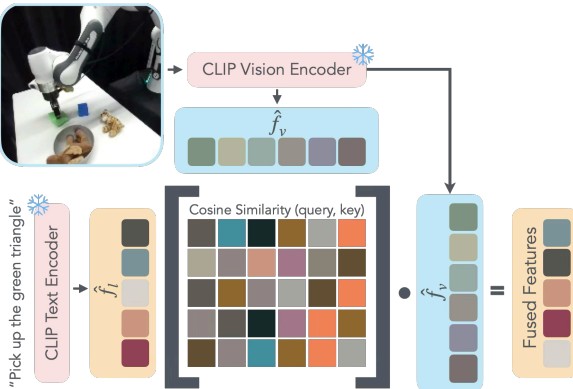

Parameter-Free Vision-Language Fusion

**Figure 3: Vision-Language Early Fusion** We calculate the similarity between the visual patch features and per-token language features, then take the softmax over the patch feature dimension. Intuitively, this give a distribution of semantic similarity over all spatial locations. We then multiply the visual patch features to retrieve the visual semantic features that correspond to each token in the sentence.

Since the language features and the visual features have different dimensions, CLIP uses a matrix per modality to project the network's output feature to the same latent dimension, denoted as $w_l$ and $w_v$ for language and vision respectively. We normalize the text and visual features for vision-language fusion. The text features are normalized using the final layer normalization: $\hat{f}_l = \text{LN}_{\text{final}}(f_l) w_l$. The visual features are normalized using the post-attention layer normalization: $\hat{f}_v = \text{LN}_{\text{post}}(f_v) w_v$. We apply L2 normalization to both text and visual features: $\hat{f}_l = \hat{f}_l / \|\hat{f}_l\|_2$ and $\hat{f}_v = \hat{f}_v / \|\hat{f}_v\|_2$ as in standard CLIP.

With the normalized features, we perform temperature-weighted attention:

$$f_{vl} = \text{softmax}(\hat{f}_l \hat{f}_v^\top / \tau) \hat{f}_v \tag{4}$$

where $\tau$ is the temperature parameter. Same as in CLIP (Radford et al., 2021), $\tau$ is learnable and is clipped between 0 and 100. The resulting feature $f_{vl} \in \mathbb{R}^{m \times d}$ are the fused vision-language tokens, where each row is a linear combination of normalized visual features $\hat{f}_v$. Intuitively, the

Simulation Scenes          Physical Scenes

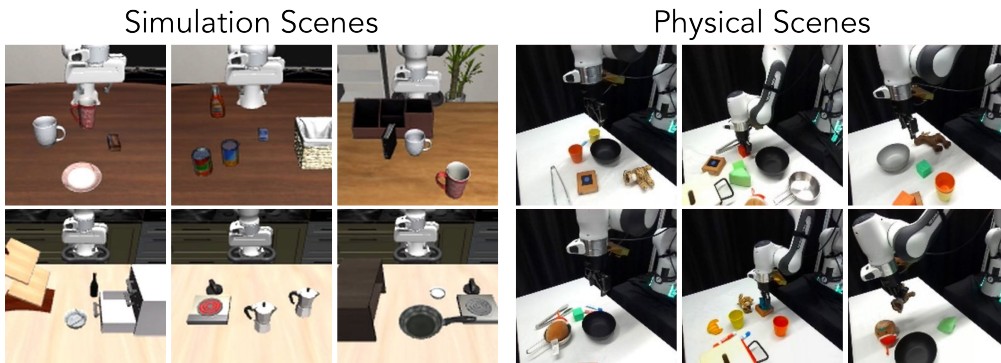

**Figure 4:** Example scenes in the simulation (left) and in the physical environments (right) using a Franka robot.

*softmax* serves as a selection function, where patch features relevant to a particular language token are selected, and a weighted average of these patches is calculated to provide cues to where the robot policy should pay attention to. A smaller $\tau$ sharpens the *softmax*, concentrating the selection on the patch with the most similar feature, while a larger $\tau$ produces a smoother, more evenly distributed selection across patches. Critically, all parameters except the $\tau$ are *frozen* throughout the training.

### 3.2 MODEL ARCHITECTURE

**Policy Network Input** We compress the fused vision-language features $f_{vl}$ into a single token for each camera. To achieve this, we apply a *learnable* cross-attention pooling operation to each camera's $f_{vl}$ to obtain a single feature $f'_{vl}$. Specifically, we use $N_q$ learnable queries $q$, and keys $k$ and values $v$ from $f_{vl}$, and compute the output using cross attention $X_{attn}(q, k, v)$. We concatenate the $N_q$ output tokens to one single token, which is $f'_{vl}$. To facilitate both early and late fusion of language features for better instruction following capabilities, we additionally employ another *learnable* cross-attention pooling on the text features $f_l$, resulting in a single text token $f'_l \in \mathbb{R}^{d_l}$. The robot's proprioceptive state is encoded through an FFN to extract an embodiment feature $f_e$. At time step $t$, we concatenate the embodiment feature $f_e$ with the perception feature $f'_l$ and $f'_{vl}$ along the channel dimension to create a single token $f_t$. This token serves as input to a policy network for action prediction.

**Policy Network and Action Head** Our policy model is a transformer consists of 4 layers and 8 heads, with a hidden dimension of 512. Fed by the combined features from the perception and embodiment, the model generates an action $a_t$. The model is trained with a context length of 12 steps. For each output token at a given timestep, we use an FFN to predict the next 12 actions. More details about our model architecture can be found in Appendix B.

**Proprioception Parametrization** We parameterize the proprioception space using a 10-dimensional representation. This includes the absolute end effector translation (x, y, z), a 6DoF rotation vector, and a continuous end-effector gripper state. The 6DoF rotation vector is derived by flattening the first two rows of the $SO(3)$ rotation matrix.

**Action Parametrization** We employ delta end effector pose as our action parameterization. At each prediction step, the model predicts $t$ actions. Given a sequence of *absolute* end effector action transforms $T_1, T_2, \cdots, T_t$ in a trajectory and the current end-effector pose $T_{ee}$, we define the relative transforms that the model needs to predict as $T_{ee}^{-1}T_1, T_{ee}^{-1}T_2, \cdots T_{ee}^{-1}T_t$. We then append the continuous absolute gripper position to each delta action. Similar to the proprioception representation, we express the delta action using the relative end effector translation and a 6DoF rotation vector, resulting in a 10-dimensional action representation.

When executing the predicted actions, we employ temporal ensembling (Zhao et al., 2023) in conjunction with receding horizon control (Chi et al., 2023). Through experimentation, we determined that an action horizon of 8 steps yields optimal performance.

## 4 EXPERIMENTS

We consider two classes of problems: language-conditioned multi-task learning and zero-shot generalization in unseen environments. For language-conditioned multi-task learning, given a multi-task setup (defined as in there are many tasks that can be performed in the same scene), the policy needs to perform the correct task corresponding to the language instruction. In the zero-shot

| Method | LIBERO-Spatial | LIBERO-Object | LIBERO-Goal | Unseen |
|---|---|---|---|---|
| EF-VLA w.o. CLIP vision | $59\% \pm 7.3\%$ | $62\% \pm 7.8\%$ | $68\% \pm 6.3\%$ | $29\% \pm 8.7\%$ |
| LF-VLA | $\mathbf{72\% \pm 9.2\%}$ | $51\% \pm 7.4\%$ | $\mathbf{76\% \pm 8.4\%}$ | $28\% \pm 11\%$ |
| EF-VLA w.o. $f_e$ | $62\% \pm 6.3\%$ | $58\% \pm 9.1\%$ | $61\% \pm 8.7\%$ | $48\% \pm 7.8\%$ |
| EF-VLA w.o. $f_l'$ | $61\% \pm 9.9\%$ | $47\% \pm 9.4\%$ | $57\% \pm 10.3\%$ | $49\% \pm 9.9\%$ |
| EF-VLA (Ours) | $71\% \pm 7.3\%$ | $\mathbf{64\% \pm 9.2\%}$ | $73\% \pm 9.4\%$ | $\mathbf{59\% \pm 7.4\%}$ |

**Table 1:** Simulation results on LIBERO. We evaluate EF-VLA and baselines on 300 trials on in-distribution tasks, and 100 trials on unseen tasks.

generalization setup, the policy is provided with a language description of an unseen task, and is asked to perform the specified task in the unseen environments. In this section, we first introduce our experimental setup to evaluate the instruction-following and visual-language alignment generalization of EF-VLA in Section 4.1. We compare EF-VLA against several baseline and ablation models in simulation and real-world in Section 4.2 and Section 4.3. In Section 4.4, we further investigate EF-VLA's capabilities by scaling up models.

## 4.1 ENVIRONMENT SETUP

**Simulation Environment** We use the LIBERO benchmark (Liu et al., 2024) for simulation evaluation. Specifically, we use LIBERO-Spatial, LIBERO-Object, LIBERO-Goal, and LIBERO-90 as the pre-training dataset, which contains 120 tasks with diverse objects, scene layouts, and language instructions. Each simulation task has 50 demonstrations. We evaluate EF-VLA's capabilities on both in-distribution tasks and unseen tasks. The in-distribution tasks are the 30 tasks in the original LIBERO-Spatial, LIBERO-Object, and LIBERO-Goal, which can evaluate the model's multi-task learning capabilities. In addition, we also construct 10 novel tasks, where we modify the language instructions and corresponding objects of 10 original LIBERO-90 tasks. For the 10 unseen tasks, we follow the same convention in LIBERO (Liu et al., 2024) about object initialization and goal configuration by defining task bddl files. Example scenes in the simulation are shown in the left column of Figure 4.

**Real Robot Environment** For real-robot evaluation, we assess all models on pick-and-place tasks with varying target objects to pick up and target placement locations. We collect a robotic dataset on multi-task scenes using a Franka robot. We consider 10 pick-and-place tasks each containing 50-80 demonstrations of human tele-operating the robot, resulting a total of 724 demonstrations. We denote this dataset as DS-PnP. We consider 10 in-distribution training tasks and 7 out-of-distribution unseen tasks for model evaluation. We consider an *unseen* combination of the target object to pick up and the target placement to place as an unseen task. The training tasks involve combinations encountered during model training, whereas the unseen tasks test the model's ability to generalize to unseen objects or scenes. Example scenes in real are shown in the right column of Figure 4.

For each experiment trial, we vary the location of the target object to pick up and introduce 2 random distractor objects, to evaluate the instruction following capability of the VLA models. In the unseen tasks, we provide the robot with novel target objects that are unseen during training, or novel combinations of target objects and target placement locations. This setup aims to evaluate both object identification and task completion ability under more challenging and previously unseen conditions. For each task (both in-distribution and unseen), we generate 10 randomized scenes, resulting in a total of 100 trials for the in-distribution training tasks and 70 trials for the unseen tasks. The robot must identify and interact with the correct object based on the provided language instruction and complete the assigned task.

The trial is terminated either when the task is completed or when a time limit is reached. The overall performance is measured by calculating the average success rate with standard error across all trials for the training and unseen tasks. The full lists of simulation and real-world environments and more experiment details can be found Appendix A.

To evaluate the model performance on task primitives other than pick and place, we additionally collect data on 3 task primitives: pouring, poking and opening/closing a drawer. For each primitive we collect around 200 demonstrations. We evaluate models on unseen tasks for these primitives. We denote the dataset consisting these 3 primitives and the pick and place primitive as DS-ALL. Details on evaluation tasks are in Appendix A.

## 4.2 EF-VLA V.S. LATE-FUSION VLA

To evaluate if the early fusion in EF-VLA can better leverage the semantic understanding capabilities of the pre-trained VLMs, we consider three baselines with late-fusion architectures, including two state-of-the-art open-sourced VLA models and one late fusion variant of EF-VLA:

1. Octo (Octo Model Team et al., 2024), an open-sourced transformer-based policy trained from scratch on 800K trajectories from the Open X-Embodiment dataset (Collaboration et al., 2024).

2. OpenVLA (Kim et al., 2024), a fine-tuned Prismatic-7B (Karamcheti et al., 2024) VLM on the Open X-Embodiment (OXE) dataset.

3. LF-VLA: a late fusion variant of EF-VLA where the text tokens, vision tokens are passed to an attention pooling layer separately to obtain independent tokens, which are then concatenated with the embodiment feature $f_e$ as the input to the transformer.

As Octo and OpenVLA are pre-trained on a real robotics dataset, we evaluate both models in the physical environments. For fair comparisons, we fine-tune Octo and OpenVLA on DS-PnP using the same amount of learning steps. The physical experiment results are reported in Table 2. We compare the performance of EF-VLA trained from scratch and EF-VLA-OXE pre-trained on the OXE dataset and fine-tuned on DS-PnP. More details about model training and architectures are in Appendix B.

In both the training and unseen tasks, Octo struggles to accurately identify the object of interest and determine the correct placement location, leading to a low success rate. We hypothesize this can be attributed to two key factors. First, Octo does not incorporate a pre-trained VLM, such as CLIP, into its network. Instead, it trains its vision encoder from scratch using a large-scale robotic dataset (OXE (Collaboration et al., 2024)), which lacks the semantic diversity found in larger vision datasets like LAION (Schuhmann et al., 2022). Second, EF-VLA applies an early-fusion strategy on CLIP's visual and text representations, which results in a stronger alignment between vision and language. This enables better visual grounding and generalization capabilities of EF-VLA to perform better on training and unseen tasks, despite being trained on a small robotic dataset. OpenVLA and LF-VLA perform similarly, which is better than Octo on training tasks, but much worse than EF-VLA. On unseen tasks, they both fail to generalize. We hypothesize this is because it's challenging for the late fusion architectures to learn generalizable vision-language connections on a small robotic dataset, while EF-VLA can utilize the early-fused vision-language features from the pre-trained VLM. EF-VLA-OXE performs better than EF-VLA on both training and unseen tasks, suggesting that EF-VLA's performance scales with more data.

We also compare LF-VLA with EF-VLA in simulation as shown in Table 1. On LIBERO-Spatial and LIBERO-Goal, LF-VLA and EF-VLA work similarly well. That's because the task semantics in LIBERO-Spatial and LIBERO-Goal can be easily distinguished. However, on LIBERO-Object, LF-VLA is worse than EF-VLA because the objects are very similar, and LF-VLA cannot accurately find the correct object to interact with. On unseen tasks, EF-VLA can outperform LF-VLA by a large margin, which is aligned with the real-world experiments.

## 4.3 ABLATIONS ON MODEL DESIGN

We consider the following ablations on the design choices of EF-VLA that are trained on DS-PnP. Full details about model training and architectures can be found in Appendix B.

1. EF-VLA w.o. $f_e$: EF-VLA without the embodiment representation $f_e$. The concatenated text token $f_l'$ and fused vision-language token $f_{lv}'$ are passed as the input to the transformer.

2. EF-VLA w.o. $f_l'$: EF-VLA without the text token $f_l'$. Only $f_{lv}'$ and $f_e$ are concatenated as the input to the transformer.

3. EF-VLA w.o. CLIP vision: EF-VLA using a small VIT to train from scratch instead of a frozen pre-trained CLIP vision encoder.

4. EF-VLA (Finetune CLIP): EF-VLA with the CLIP initialized from the pre-trained weight and fine-tuned end to end on the robotic dataset.

| Method | Training Tasks | Unseen Tasks |
|---|---|---|
| Finetuned Octo | $15\% \pm 3.4\%$ | $12\% \pm 3.6\%$ |
| EF-VLA w.o. CLIP vision | $17\% \pm 2.9\%$ | $11\% \pm 2.5\%$ |
| Finetuned OpenVLA | $30\% \pm 3.9\%$ | $9\% \pm 3.1\%$ |
| LF-VLA | $29\% \pm 3.7\%$ | $4\% \pm 1.6\%$ |
| EF-VLA (Finetune CLIP) | $26\% \pm 4.0\%$ | $15\% \pm 3.9\%$ |
| EF-VLA w.o. $f_e$ | $40\% \pm 4.0\%$ | $29\% \pm 4.3\%$ |
| EF-VLA w.o. $f_l'$ | $57\% \pm 4.4\%$ | $53\% \pm 4.6\%$ |
| EF-VLA (Ours) | $68\% \pm 4.3\%$ | $62\% \pm 4.2\%$ |
| **EF-VLA-OXE (Ours)** | **$72\% \pm 3.9\%$** | **$73\% \pm 2.8\%$** |

**Table 2:** Physical results on 100 trials on in distribution training tasks and 70 trials on unseen tasks. EF-VLA achieves similar success rate on the in distribution training tasks and unseen tasks, significantly outperforming the baselines, highlighting the benefits of using early fusion and a frozen pre-trained VLM.

**Simulation Results** Table 1 presents the simulation results of EF-VLA and other ablations. On the in-distribution tasks, EF-VLA w.o. CLIP vision and EF-VLA work similarly well given sufficient demonstrations, but EF-VLA w.o. CLIP vision drops 51% on unseen tasks, which shows the benefits of using a pre-trained VLM for better generalization capabilities.

The performance of EF-VLA w.o. $f_e$ drops about 10% on both the in-distribution and unseen pick and place tasks, indicating that $f_e$ is beneficial for task completion as it provides explicit spatial information of the robot. EF-VLA w.o. $f_l'$ is also noticeably worse, especially for LIBERO-Object and LIBERO-Goal. We hypothesize this is due to the object are not very realistic in simulation, so the early fusion in CLIP's may highlight multiple objects or wrong objects. $f_l'$ can provide complementary information for the transformer to interact with the correct objects.

**Real-world Results** Physical results in Table 2 suggest that the performance on both the training tasks and the unseen tasks drop significantly for the ablations compared to EF-VLA.

Similar to the simulation results, the performance of EF-VLA w.o. $f_e$ drops 28% on the training tasks and 33% on the unseen tasks, indicating that $f_e$ is vital for task completion and generalization, likely because it provides a physical grounding for decision-making. Without $f_e$, the model's understanding of embodied features, possibly linked to the spatial or physical aspects of the task, is severely impaired. EF-VLA w.o. $f_l'$ experiences a performance drop of around 10% on both training and unseen tasks but maintain a decent performance, suggesting that $f_l'$ provides complementary information that may help in more nuanced task understanding, aligned with the simulation results.

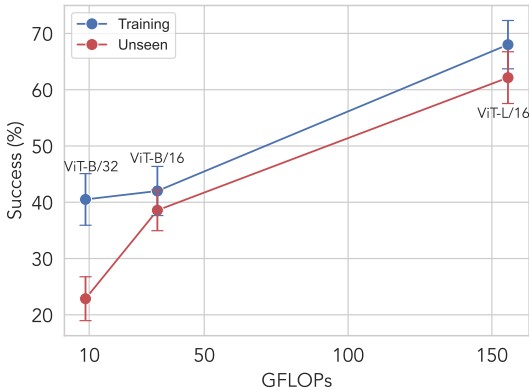

**Figure 5:** We evaluate EF-VLA's performance with improved vision language features by scaling CLIP. In particular, we train EF-VLA with three CLIP variants with increasing FLOPs: ViT-B/32, ViT-B/16, and ViT-L/16. We report the task performance vs. the inference FLOPs per image on training and unseen tasks. The results suggest that the EF-VLA can benefit from scaling up vision-language model.

While EF-VLA w.o. CLIP vision shows decent performance on in distribution tasks in simulation experiments, it has a significant performance drop of more than 50% on the training and unseen tasks in physical experiments. The results of EF-VLA w.o. CLIP vision is similar to Octo which also trains a vision encoder from scratch on the robotics dataset. This suggests that pre-trained VLM provides more robust and transferable visual representations. Training a vision encoder from scratch can result in poor performance, as it lacks the generalization capabilities learned from large-scale pre-training.

OpenVLA suggests that fine-tuning the vision encoder of the pre-trained VLM on the robotics dataset is crucial for improving the performance of a late fusion VLA. However, we hypothesize that fine-tuning a pre-trained VLM can diminish the general vision-language understanding capabilities of a VLM obtained through pre-training on internet-scale vision language datasets. EF-VLA (Finetune CLIP) shows worse performance on both the training tasks and the unseen tasks. This may be

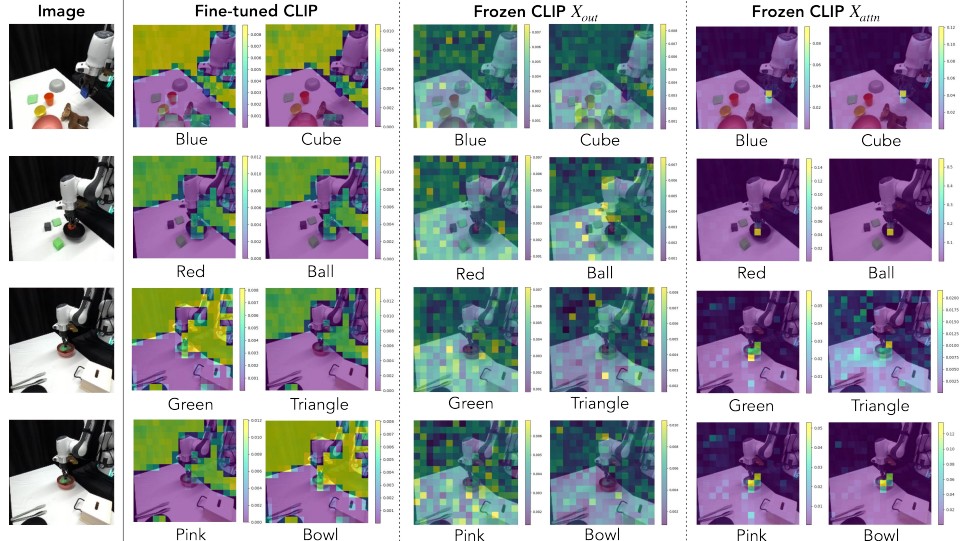

**Figure 6:** Examples of attention maps for CLIP fine-tuned with VLA (left) and frozen CLIP's output ($X_{out}$) (middle) and frozen CLIP's attention features ($X_{attn}$) (right). The first column shows the side view observation and the text query is below each attention map. Fine-tune CLIP pays attention to the background and the frozen CLIP's output ($X_{out}$) is noisy. In contrast, the frozen CLIP ($X_{attn}$) pays attention to the correct object associated with the text query. These examples indicate that fine-tuning CLIP on robotic datasets can degrade the performance of the pre-trained CLIP, especially when the robotics dataset is small. It also highlights the benefits of using $X_{attn}$ for fused vision-language features.

attributed to that a fine-tuned CLIP vision encoder is easier to over-fit on the training data and that a fine-tuned CLIP vision encoder has a degraded vision-language understanding capabilities. The large performance discrepancy between the training tasks and unseen tasks of OpenVLA and EF-VLA (Finetune CLIP) implies a worse vision-langauge generalization ability, showing the benefits of EF-VLA for retaining the vision-language features from a frozen pre-trained VLM. It's worth noting that both early fusion of the vision-language features and the frozen VLM is crucial for learning a VLA that can generalize to unseen tasks, as shown by the worse performance of LF-VLA with a frozen VLM, EF-VLA (Finetune CLIP) that has a fine-tuned VLM and OpenVLA that is a late fusion model with fine-tuned VLM.

## 4.4 SCALING UP VISION-LANGUAGE MODEL

The semantic understanding capability of VLMs scales with model capacity and compute (Radford et al., 2021). To understand whether EF-VLA can leverage the advances of pre-trained VLMs, we evaluate its performance when trained on DS-PnP with three CLIP models with increasing floating point operations per model forward pass: ViT-B/32, ViT-B/16, and ViT-L/14. The task success and the inference FLOPs per image are provided in Figure 5. We observe significant improvements of EF-VLA when scaling up CLIP for training and unseen tasks, indicating that EF-VLA is a scalable approach that effectively utilizes pre-trained vision language models for downstream robotics tasks.

## 4.5 GENERALIZATION PERFORMANCE ON MORE TASK PRIMITIVES

We compare the performance of Octo and OpenVLA finetuned on DS-ALL and EF-VLA pretrained on OXE and fine-tuned on DS-ALL, denoted as EF-VLA-OXE. As there are more primitives, we also consider a deeper and wider EF-VLA model (details in Table 6), denoted as EF-VLA-OXE-L. For a fair comparison, we extended the context history length of Octo to 10 (Octo cannot exceed a context length of 10 due to its inherent design constraints) and matched its action prediction horizon to ours. As OpenVLA has many tokens per timestep, its context length cannot be extended and we use its default context length. All models are evaluated on unseen tasks for each primitive, with 10 trials for each task. Results are shown in Table 3, where the performance of Octo, OpenVLA and EF-VLA-OXE on the pick and place task all drop, showing the difficulty of multi-primitive learning. Notably, both Octo and OpenVLA fail to complete any unseen tasks for the pouring, drawer and poking tasks, likely due to a relatively small amount of demonstrations for each primitive. Both

EF-VLA-OXE and EF-VLA-OXE-L can achieve high success rate on all four primitives on the same amount of demonstrations, indicating that using fused vision language features from a pre-trained VLM can increase the data efficiency and enhancing the generalization ability. EF-VLA-OXE-L outperforms EF-VLA-OXE on average, indicating that EF-VLA can scale with model size.

| Method | Pouring | Drawer | Poking | Pick and Place | Average |
|--------|---------|--------|--------|----------------|---------|
| Finetuned Octo (long) | 0% | 0% | 0% | 5% | $4\% \pm 1.2\%$ |
| Finetuned OpenVLA | 0% | 0% | 0% | 1% | $0.6\% \pm 0.5\%$ |
| EF-VLA-OXE | 60% | 65% | **93%** | 66% | $70\% \pm 3.6\%$ |
| EF-VLA-OXE-L | **77%** | **75%** | **93%** | **75%** | $\mathbf{77\% \pm 3.3\%}$ |

**Table 3:** Physical results on 150 trials on unseen tasks for 4 different primitives. EF-VLA achieves the highest success rate all unseen tasks, significantly outperforming the baselines.

### 4.6 VISION-LANGUAGE ATTENTION VISUALIZATION

In Figure 6, we visualize the cosine similarity between the output of the CLIP ViT-L/16 encoder and the per-token text features in three different settings: (1) fine-tuning the encoder, (2) a frozen CLIP's output features ($X_{out}$), and (3) a frozen CLIP's last attention block's feature ($X_{attn}$) as described in Section 3.1. A more in-depth analysis and more examples can be found in Appendix C.

In the finetuning v.s. frozen CLIP ($X_{attn}$) comparison, fine-tuning EF-VLA's CLIP results in overfitting to foreground-background separation, causing it to lose zero-shot object detection ability. This limits the model's ability to highlight the correct object, leading to a significant drop in task success rates (26% vs 68% for training tasks and 15 vs 62% for unseen tasks). Conversely, a frozen CLIP ($X_{attn}$) preserves object detection capabilities, providing better downstream performance.

In the Vanilla CLIP output ($X_{out}$) v.s. ClearCLIP output ($X_{attn}$) comparison, CLIP produces noisy features, degrading vision-language alignment and making object localization harder. By using the attention output ($X_{attn}$) as in ClearCLIP (Lan et al., 2024) instead of the final feature map, EF-VLA can localize objects more accurately without fine-tuning or additional parameters.

## 5 LIMITATIONS AND CONCLUSIONS

While EF-VLA demonstrates improved task completion rates compared to existing VLAs, it still faces several limitations. One significant challenge is scaling across different morphologies, particularly those that cannot be easily parameterized by SE(3) transforms (i.e. robot multi-finger hand). This limitation restricts the model's adaptability to a wider range of robotic platforms and task types. Furthermore, this study has not extensively explored how this method scales with larger datasets or more complex tasks. This leaves open questions about the model's performance and generalization capabilities in more challenging scene configurations, which could be an important area for future research and potential improvement of the EF-VLA approach.

In summary, we present EF-VLA, a vision-language-action model that implements early fusion between vision and language features. This is achieved by utilizing a pre-trained vision-language model and an early fusion method to extract task-relevant semantic information. The experimental results demonstrate that this early fusion approach enables effective multi-task learning with few demonstrations and facilitates extrapolation to unseen objects and environment configurations. The results further suggest that EF-VLA has a higher task success rate in handling unseen scenes with distractor objects than the existing state-of-the-art VLAs.

## 6 REPRODUCIBILITY STATEMENT

The simulation benchmarks (Liu et al., 2024) and the real robot setup (Khazatsky et al., 2024) are already open-sourced. The model's hyperparameters and implementation detail are listed in Appendix B. We commit to releasing all of the code, data, and models to accompany the paper.

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

# A    ENVIRONMENT SETUP

## A.1    SIMULATION TASKS

For the training tasks, we use the original tasks in LIBERO-Goal, LIBERO-Spatial, and LIBERO-Object. We also build unseen evaluation tasks based on 10 original LIBERO-90 tasks, by changing language instructions and target object color and type in the task bddl files. The 10 unseen tasks are listed in Table 4.

| Changes | Unseen |
|---|---|
| object type | Put the moka pot in the bottom drawer of the cabinet |
| object type | Put the moka pot on the wine rack |
| object type | Pick up the ketchup and put it in the basket |
| object type | Pick up the ketchup on the plate |
| object type | Pick up the bottle and put it in the tray |
| object color | Put the black bowl on top of the cabinet |
| object color | Put the black bowl on the plate |
| object color | Put the red mug to the right of the plate |
| object color | Put the yellow and white mug in the front of the red mug |
| object color | Put the red mug to the front of the moka |

**Table 4:** The 10 in-distribution tasks and 7 unseen tasks we used in our real-world setting.

## A.2    REAL-WORLD TASKS

The full list of tasks for our real-world evaluation is provided in Table 5.

| In-Distribution | Unseen |
|---|---|
| Put potato in pot to black bowl | Put yellow cube in black bowl |
| Pickup potato | Pick up radish and place it in grey bowl |
| Pick up and place deer in grey bowl | Put blue bear in pink bowl |
| Pick up green triangle | Put yellow cube in grey bowl |
| Put tiger to black bowl | Put apple with a green leaf in black bowl |
| Put red cube into black bowl | Pick up blue sponge and place it in steel pot |
| Put blue cube into grey bowl | Pick up black dog and place it in the pink bowl |
| Put the red ball in black bowl | |
| Put green triangle into pink bowl | |
| Put blue cube in pink bowl | |
| Poke a wooden block | Poke the radish |
| Poke a tiger | Poke the gray dog |
| Poke a green triangle | Poke the pink bowl |
| Poke a gray bowl | |
| Pour from the brown cup to the gray bowl | Pour from the orange cup to the black bowl |
| Pour from the blue cup to the pink bowl | Pour from the blue cup to the black bowl |
| Pour from the yellow cup to the black bowl | Pour from the brown cup to the pink bowl |
| Open the drawer | Open the drawer with a tiger on top |
| Close the drawer | Close the drawer with a red cube inside |

**Table 5:** The 10 in-distribution tasks and 7 unseen tasks we used in our real-world setting.

For each experiment trial of poking and pouring, we vary the location of the target object to manipulate and introduce 2 or 3 random distractor objects. For drawer, we vary the location of the drawer on each trial. Similar to the pick and place primitive, for each task, we generate 10 randomized scenes.

Each trial is scored based on the robot's performance in completing the task. For the pick and place primitive, a score of **0.5** is awarded if the robot successfully picks up the correct target object, and a score of **1** is given if the robot not only picks up the correct object but also places it in the correct location as specified by the instruction. If the robot fails to pick up the target object or picks up a

distractor object, a score of **0** is recorded. For other primitives, a score of **1** is recorded if the task is completed, otherwise a score of **0** is given.

For all models other than the OpenVLA, each trial is allowed a maximum of 30 seconds to complete. As OpenVLA is a large 7B model wit a lower inference speed, we give it a time limit of 60 seconds to complete a task.

# B    MODEL AND TRAINING DETAILS

## B.1    MODEL ARCHITECTURE FOR EF-VLA AND BASELINES

The details of our model parameters can be found in Table 6. All the baselines share the same hyper-parameters with EF-VLA. For EF-VLA w.o. CLIP Vision, we use a ViT Encoder based on the implementation of `https://github.com/google-research/vision_transformer` with a ViT-Ti/16 configuration with half of the number of attention layers. For EF-VLA w.o. $f_e$ and EF-VLA w.o. $f'_l$, we use the same model configuration but only remove the corresponding attention pooling layers. We incorporate action chunking into OpenVLA by asking it to predict the next 16 actions, which performs better than vanilla OpenVLA which predicts only the next step. For Octo, we use the official Hugging Face Checkpoint at `hf://rail-berkeley/octo-small-1.5` which is in a comparable size with our model. During inference, we cache the CLIP feature outputs. This enables the ViT-L/14 EF-VLA model to perform inference at $> 15Hz$ on a single NVIDIA 3090Ti, allowing real-time control.

| Hyperparameter | Value |
|---|---|
| CLIP Model | ViT-L/14 |
| # Pooling Readouts | 4 |
| # Pooling Attention Heads | 8 |
| # Pooling Attention Blocks | 2 |
| # Text-Pooling Output Dimension | 128 |
| # Image-Pooling Output Dimension | 512 |
| # Proprio-Pooling Output Dimension | 64 |
| *Causal Transformer Parameters:* | |
| # Attention Blocks | 4 (8) |
| # Attention Heads | 8 |
| # Latent Dimension | 512 (768) |
| # Context Length | 12 |
| # Action Prediction Horizon | 12 |

**Table 6:** Hyperparameters for EF-VLA model architecture. Values in the parenthesis shows the hyperparameters for a larger and wider EF-VLA.

## B.2    TRAINING HYPER-PARAMETERS

We use the AdamW optimizer with a cosine learning rate decay schedule and linear learning rate warm-up. We list training hyperparameters in Table 7. All these hyper-parameters are shared between real-world and simulation. All the models are trained on 4 NVIDIA A100 80GB GPUs.

# C    VISION-LANGUAGE ATTENTION VISUALIZATION

To provide further motivations for why using $X_{out}$ (per (Lan et al., 2024)) instead of the output feature map of CLIP, we compare the cosine similarity for each of these options respectively. Similar to what ClearCLIP has noted, after adding residual connection and the final FFN, the features become noisy and worsen the alignment between language and visual features. The noisy attention map makes it challenging for the model to identify the correct features directly from the feature map, which makes it necessary for existing VLA (i.e. OpenVLA (OpenAI, 2024)) to fine-tune the CLIP vision encoder. In comparison, by using $X_{attn}$, object localization becomes an easier task in EF-VLA: we can extract the location of the object by getting the *softmax* across the attention map without using any parameters (see Figure 3). More attention map examples on Open-X dataset are in Figure 7.

| Hyperparameter | Value |
|---|---|
| Learning Rate | 3e-4 |
| Warmup Steps | 2000 |
| Weight Decay | 0.01 |
| Learning Rate Scheduler | cosine |
| Gradient Clip Threshold | 1 |
| Batch Size | 64 |
| Total Gradient Steps | 40000 (60000) |
| Image Resolution | $224 \times 224$ |
| Random Resized Ratio | [0.9, 1.1] |
| Random Brightness | 0.2 |
| Random Contrast | [0.8, 1.2] |
| Random Saturation | [0.8, 1.2] |
| Random Hue | 0.1 |

**Table 7:** Hyperparameters used for training (pre-training on OXE).

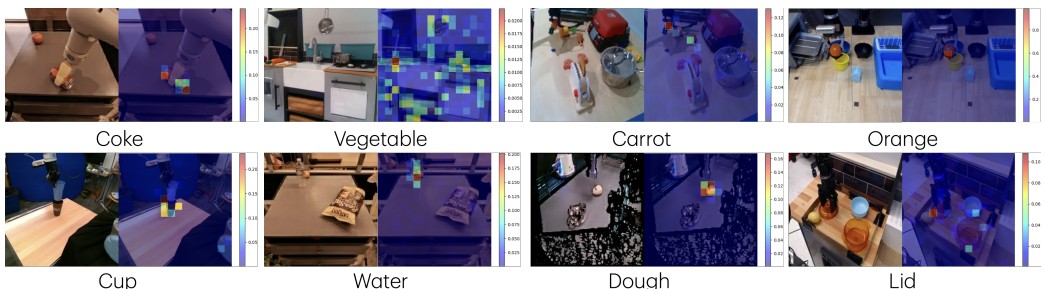

Coke    Vegetable    Carrot    Orange

Cup    Water    Dough    Lid

**Figure 7:** Examples of attention maps of frozen CLIP's attention features (Xattn) on Open-X dataset. The bottom texts are the corresponding text tokens.

It may initially seem unexpected that this type of visualization is reasonable. However, this can be explained by the fact that LayerNorm operates independently of the patch dimension, as it normalizes along the channel dimension. When combined with the vision-alignment weight matrix $w_v$, the operation $\hat{f}_v = \text{LN}_{\text{post}}(f_v)w_v$ remains linear. Therefore we can linearize the final attention block:

$$\hat{f}_v = \text{LN}_{\text{post}}(X_{out})w_v \tag{1}$$
$$= \text{LN}_{\text{post}}(X_{res} + X_{attn} + \text{FFN}(\text{LN}(X_{sum})))w_v \tag{2}$$
$$= \text{LN}_{\text{post}}(X_{res})w_v + \text{LN}_{\text{post}}(X_{attn})w_v + \text{LN}_{\text{post}}(\text{FFN}(\text{LN}(X_{sum})))w_v \tag{3}$$

For ClearCLIP, or Frozen CLIP $X_{attn}$, we are visualizing the $\text{LN}_{\text{post}}(X_{attn})w_v$ term.

## D   MORE ABLATIONS

We consider another 2 ablations of EF-VLA.

1. LF-VLA (CLS): another late fusion variant of EF-VLA that utilizes the CLS token rather than cross-attention pooling on all the patch tokens.

2. EF-VLA (xattn): EF-VLA using standard cross attention pooling between the text tokens $f_l$ and the vision tokens $f_v$ to obtain the fused vision language features $f'_{lv}$ instead of doing patch-wise alignment as in Eq.( 4).

From Table 8, both LF-VLA (cls) and EF-VLA (xAttention) fails to generalize to unseen tasks, highlighting the benefits of using ClearCLIP to obtain task related vision features as the fused vision language features.

| Method | LF-VLA (CLS) | EF-VLA (xattn) | EF-VLA |
|---|---|---|---|
| Success Rate | $6\% \pm 0.8\%$ | $2\% \pm 0.5\%$ | $62\% \pm 4.2\%$ |

**Table 8:** Physical results on 70 trials on unseen tasks for other variants of EF-VLA.

