# OpenReview forum: "Early Fusion Helps Vision Language Action Models Generalize Better"
_ICLR.cc/2025/Conference — Submitted to ICLR 2025_

### Official Review · Reviewer_8sZj · 2024-10-28

**Soundness:** 1
**Presentation:** 2
**Contribution:** 1
**Rating:** 1
**Confidence:** 5

**Summary:**

This paper proposes "Early Fusion VLA" (EF-VLA) to address token fusion from different modalities in a modern vision-language-action model, i.e., a policy network that takes vision-language inputs and predicts robot actions. EF-VLA is motivated by 1. leveraging existing vision-language alignment in pre-trained models (like CLIP) instead of re-learning it at the policy learning stage (late fusion), and 2. providing fine-grained visual features for policy learning. The proposed EF-VLA uses CLIP as the vision encoder and follows the ClearCLIP way to use per-patch features for language alignment. The author evaluates EF-VLA only in a simulated environment and compares it against OCTO and OpenVLA in real-world experiments. The results show that EF-VLA improves.

**Strengths:**

- Studying early or late fusion is meaningful for robot learning use case.

- Real-world experiments are highly appreciated.

**Weaknesses:**

- The authors claim that VLMs and VLA models using similar architectures are in the late-fusion style (L142-144). However, I hold a different opinion. Let's take examples of LLaVA and OpenVLA. In LLaVA, the visual encoder is CLIP. The visual tokens are generated by CLIP, mapped to language space, and processed by the LLM part. The LLM part is visually very large, for example, 8B, 34B, and even 72B / 110B (recent LLaVA-NeXT). Comparing the LLM parameters to CLIP parameters (~300M for CLIP-ViT-L), the visual tokens still go through many more parameters than the encoder part. OpenVLA uses DINOv2 and SigLIP plus a Llama-7B backbone, in a similar way that many more parameters are used for modality fusion. Therefore, I don't believe it's true that many existing VLMs and VLA models are late-fusion.

- The presentation is extremely unclear. See some clarification questions below.
  * L176 Eq. (2) What are $X_{res}$ and $X_{attn}$? They are not defined.

  * Is Eq. (1) overloaded in L175 and L212? Also, the equation in L212 does seem to be standard (more like pseudo-code)

  * As this paper follows ClearCLIP, it seems that this paper is directly taking Eq. (1)-(3) from the original ClearCLIP paper, but poorly explains the meaning.

  * What's the cosine similarity in Figure 3? I can not find anything in the main text that describes it.

- What's the reason for using final LN for text features but post-attention LN for visual features? The choice needs justification. Following this, what's the meaning of another L2 normalization descript in L207? Are they referring to the same operation?

- One of the main motivations is to provide dense visual features to policy learning (L84-86). However, the aligned vision-language tokens $f_{vl}$ are further compressed into one token. The compression is done by cross-attention pooling using the original text feature. So
  1. Why do we need to do the first per-patch alignment and then compress it again? Can't we directly do the final pooling?

  2. The author needs to justify why a single token can provide dense visual features to the policy learning framework, with only light-weight learnable parameters in between (L227)

  3. The cross-attention pooling is not described (L227)

- The real-world evaluation tasks are all pick-and-place tasks. This is a very limited setting. And I'm concerned that the advantage of this model only preserves this limited task scope, which is unfair to other VLA models like OCTO and OpenVLA.

- Significant architectural difference between EF-VLA and baselines. The policy part of EF-VLA is trained from scratch from my understanding. However, authors use it to compare against fine-tuned OCTO and OpenVLA models. This does not make sense since the model sizes could significantly affect the result given the relatively small amount of real-world data, compared to the large pre-trained set that OCTO and OpenVLA used. EF-VLA's policy head only contains 4 Attention blocks, each with 8 attention heads and the latent dimension is 512. This is much smaller than OCTO and OpenVLA. Another point is that EF-VLA takes inputs from multiple steps (and also outputs multiple steps) but the others don't. The performance gain could come from this aspect but the author didn't experiment thoroughly. The better way to study early fusion is by modifying the original OCTO and OpenVLA and re-training them. I understand the potential effort on it, but there is no other way to clearly justify early fusion is better and generally applicable.

- The proposed particular method highly depends on CLIP. The problem is how this approach can be generalized to other vision models and whether other vision encoders will show similar early-fusion benefits. In its current form, the approach is highly limited by CLIP. The authors are highly encouraged to explore how other pre-trained vision encoders for general purposes or specifically for robot learning will fit in this paradigm. Also, studying the way of mixing two visual representations like in OpenVLA is interesting.

- The technical contribution is minor considering the existing ClearCLIP.

- Other baselines are not evaluated in simulated environments. Only one simulated environment is used.

**Questions:**

Please find them in the weaknesses part.

---

> ### Author Response · Authors · 2024-11-21
> **Response to reviewer 8sZj (Part 1)**
>
> We appreciate the reviewer for taking the time to review our paper and give constructive comments. We emphasize the contribution of our work and clarify some commonly raised questions in our general response. Here we address your individual questions as follows.
> #### 1. Many VLAs are not Late-fusion
> Please refer to our general response for the definition of late-fusion.
> #### 2. Unclear notations
> * Lines 175-178 are standard formulations of self-attention machinism, which employs two residual additions at Lines 177 and 178. $X_{attn}$ is the output after the attention operation, $X_{res}$ is the identical to the origibal $X$. We replaced the notation for $X_{res}$ with $X$ in the revised paper.
> * Thank you for pointing this out. We changed Line 212 to Eq. (4).
> * Eq. (1)-(3) follow a standard implementation of attention machinism. We use the same notation as in ClearCLIP, but it's also a very general formulation as proposed in [1]. We have explained the meaning of Proj, LN, and FFN in Line 179.
> * In Line 212, the cosine similarity is computed as $\hat{f}_l\hat{f}_v^T$, where $\hat{f}_l$ and $\hat{f}_v$ are normalized feature vectors as described in Line 208.
>
> [1] Vaswani et al, Attention is all you need.
>
> #### 3.1 "What's the reason for using final LN for text features but post-attention LN for visual features? The choice needs a justification"
> * A standard attention block will perform operations like Line 175-178, eq (1)-(3). Using the final output of an attention block is **default**, which reflects the training data flow. All of our attention modules, except for the ClearCLIP component, adhere to this standard, as the attention modules in other VLA models do (e.g., Octo, OpenVLA). In other words, the final attention outputs are used consistently for our text features, proprioceptive state features, policy transformer outputs, and all our baselines.
> * Following the findings of ClearCLIP, we observed that using the final LN cannot give us detailed vision-language alignment representations (as shown in Figure 6), but the post-attention LN of CLIP can offer us clean correspondence between patch and text. We then use this strategy to extract clean and fine-grained vision-language representations.
> * We provide both qualitative justification (Figure 6) and quantitative justification (Table 1 and 2) for this choice.
> #### 3.2 "Following this, what's the meaning of another L2 normalization descript in L207? Are they referring to the same operation?"
> The L2 operation is used to normalize the features to compute the cosine similarity, which is also adopted by the origial CLIP model. As shown in Line 207, they are both L2 normalization.
> #### 4.1 Why first per-patch alignment and then compress it again. Can't we directly do the final pooling?
> We first perform per-patch alignment to obtain patch-level vision-language representations. However, as illustrated in Figure 6, many patch tokens correspond to background elements or distractors, which introduce redundant or irrelevant information for the policy. The policy must learn to compress these redundant tokens to extract meaningful visual information effectively, in order to generalize to unseen distractors and backgrounds.
>
> **Final pooling**: Our LF-VLA baseline is directly doing the final pooling over vision and text features to get $f'_v$ and $f'_l$, and performs significantly worse than our method as shown in Table 2.
>
>
> **Direct cross attention without per-patch alignment (ClearCLIP)**: We also added another early fusion baseline to justify our design choice. Instead of using ClearCLIP to get $f\'\_{vl}$, we can use language tokens to generate queries, and perform cross attention to obtain $f\'\_{vl}$. We call this baseline EF-VLA (xattn). We compare EF-VLA (xattn) with EF-VLA on 7 held-out tasks, and it performs significantly worse than the original EF-VLA. This demonstrates the importance of early per-patch alignment (ClearCLIP) against directly performing various pooling strategies.
>
> | Task | EF-VLA | EF-VLA (xattn) |
> |-|-|-|
> | yellowcube-in-black-bowl| 0.6 | 0.15 |
> | yellow-cube-in-grey-bowl| 0.7 | 0.05 |
> | blue-bear-in-pink-bowl| 0.6 | 0 |
> | reddish-in-grey-bowl| 0.55 | 0.1 |
> | black-dog-in-pink-bowl| 0.8 | 0.15 |
> | blue-sponge-in-pot| 0.4 | 0 |
> | apple-in-black-bowl| 0.7 | 0 |
>
> #### 4.2 Why a single token can provide dense visual features to the policy learning framework.
> We want to convey that the vision-language correspondence is localized at the patch level, which is better described using the word **fine-grained** rather than **dense**. We updated the term in the revised paper. As shown in Figure 6, the correspondence is notably **clean, sparse, and fine-grained**.
>
> Leveraging this fine-grained and sparse correspondence, we can represent an image using $N$ $k$-dimensional tokens ($N=4, k=64$ in the paper). These tokens are then concatenated to form a single $N*k=256$-dim token.

---

> > ### Comment · Reviewer_8sZj · 2024-11-24
> >
> > 1. Please find my comments directly under your general response.
> >
> > 2. Thank you for the explanation.
> >
> > 3. Examples are in environments with clean backgrounds which is hard for me to justify. What's the case in cluttered/complex/natural backgrounds? What's the case if objects are in the same or similar color/shape/entity?
> >
> > 4.
> >
> >    - Thanks for sharing the results of this xattn baseline.
> >
> >    - The explanations are clear now. I suggest adding them to the figure for the best clarity. How are N and k chosen? How do they affect the capability of following language instructions, vision-language patch alignment, and generalizing to unseen tasks? Is the concatenation the only way to do this fusion?

---

> ### Author Response · Authors · 2024-11-21
> **Response to reviewer 8sZj (Part 2)**
>
> #### 4.3 Cross-attention pooling.
> To reduce the number of tokens, we use $N$ learnable queries $q$, and keys $k$ and values $v$ from $f_{vl}$, and compute the output using cross attention $X_{attn}(q, k, v)$. We concatenate the $N$ output tokens to one single token, which is $f'_{vl}$. We revised the paper (L226) to make it clear.
>
> #### 5. All pick-and-place tasks, unfair to OpenVLA and Octo
> We collected new human demonstrations, trained and evaluated models for three new primitives: poking an object, pouring from one cup to a container, and opening or closing drawers. For each primitive, we collect around 200 demonstrations. For more details on the tasks for each primitive, please refer to Appendix Section A in the updated manuscript.
>
> We then evaluate the finetuned models on 15 unseen tasks (10 trials per task) for the primitives and report the performance in the following table (also section 4.5):
>
> | Method               | Pouring | Drawer | Poking | Pick and Place | Average       |
> |----------------------|---------|--------|--------|----------------|---------------|
> | Finetuned Octo (long)    | 0%      | 0%     | 0%     | 5%             | 4% ± 1.2%     |
> | Finetuned OpenVLA   | 0%      | 0%     | 0%     | 1%             | 0.6% ± 0.5%   |
> | Finetuned EF-VLA-OXE         | 60%     | 65%    | 93%    | 66%            | 70% ± 3.6%    |
>
> Notably, both Octo and OpenVLA fail to complete any unseen tasks for the pouring, drawer, and poking tasks, likely due to a relatively small amount of demonstrations for each primitive. EF-VLA pre-trained on OXE (the pretraining dataset for Octo and OpenVLA) achieves a high success rate on all four primitives on the same amount of demonstrations, indicating that using fused vision language features from a pre-trained VLM can increase the data efficiency and enhance the generalization ability.
>
> #### 6. Significant architectural difference between EF-VLA and baselines
> * EF-VLA can potentially benefit from a longer context length and action prediction horizon, which could create an unfair comparison with the baselines. To address this, we extended the context history length of Octo to 10 and matched its action prediction horizon to ours. It is important to note that Octo cannot exceed a context length of 10 due to its inherent design constraints (to match the design of its pretraining). OpenVLA has many tokens per timestep, so its context length cannot be extended to 10, so we still use its default context length. We report the performance in the above table in Answer 5.
>
> * Although EF-VLA has fewer parameters compared to Octo and OpenVLA, we both trained them until convergence on the same datasets. To provide further illustration of EF-VLA's scalability, we also enlarged EF-VLA to have 8 blocks with latent dimension 768, which matches Octo. We then evaluate the EF-VLA and Large EF-VLA on 15 unseen tasks (10 trials per task) for the primitives and report the performance in the following table. We can observe that EF-VLA's performance also scales with model size, with a consistent performance advantage against Octo and OpenVLA.
>
> | Method               | Pouring | Drawer | Poking | Pick and Place | Average       |
> |----------------------|---------|--------|--------|----------------|---------------|
> | EF-VLA       | 60%      | 65%     | 93%     | 66%             | 70% ± 3.6%     |
> | Large EF-VLA          | 77%     | 75%    | 93%    | 75%            | 77% ± 3.3%    |

---

> > ### Comment · Reviewer_8sZj · 2024-11-24
> >
> > 4.
> >
> >    - Please also add them to the figure.
> >
> >
> > 5. Since the paper aims at generalization capability, why didn't directly evaluate it in zero-shot settings? I'm also concerned about the success rates of Octo and OpenVLA as they reported good generalization capability. How can we know your implementation is correct?
> >
> > 6. Thanks for the results!
> >
> >    - How can I know your long-horizon extension is correct? What's the result of the original Octo without extension?
> >
> >    - Need a case that matches OpenVLA.

---

> > > ### Public Comment · ~Yide_Shentu2 · 2024-11-28
> > > **Interesting Observations on Octo, and interesting ask about the correctness of the code.**
> > >
> > > https://bsky.app/profile/cpaxton.bsky.social/post/3lbwki5ffts27
> > >
> > > If you're concerned about the success rates of Octo and OpenVLA, I recommend checking out this discussion. Considering this, the results presented here seem quite reasonable in comparison.

---

> ### Author Response · Authors · 2024-11-21
> **Response to reviewer 8sZj (Part 3)**
>
> #### 7. The proposed particular method highly depends on CLIP; The technical contribution is minor considering the existing ClearCLIP.
>
> We respectfully disagree with such characterization. While ClearCLIP demonstrates improvements in fine-grained vision-language alignment for open-vocabulary segmentation, this work addresses a fundamentally different domain and problem: leveraging this alignment for downstream Vision-Language-Action (VLA) tasks, particularly in robotics.
>
> More importantly, the objective of this paper is to emphasize that when designing model architecture for vision language action models, we can consider ways to better utilize the existing alignment that is present in CLIP (i.e. select only the language relevant visual features as the input to the policy network), instead of re-aligning the visual representation to a language model.
>
> Thus, the experimental impact is orthogonal to that of ClearCLIP. We provide evaluations demonstrating that our approach significantly outperforms state-of-the-art late fusion methods (e.g., OpenVLA) and other baselines in both simulation and real-world tasks. These results highlight the impact of the architectural changes in the robotics context, which is distinct from the goals of ClearCLIP.
>
> By addressing these novel challenges and achieving state-of-the-art performance in VLA tasks, our work represents a substantial extension of ClearCLIP’s principles into a new and impactful domain. We hope the reviewers can recognize the broader implications and technical contributions of our approach within this context.
>
> #### 8. Other baselines are not evaluated in simulated environments. Only one simulated environment is used.
>
> Both the original Octo and OpenVLA didn't report any simulation results, so we didn't include their simulation performance in our paper. Given they are pre-trained on real-world robotics dataset, and OpenVLA also adopt pre-trained vision encoders, their real-world performance can be more convincing to be compared with. We mainly use simulation environments to conduct ablation studies. We are working on incorporating more simulation benchmarks, and will include the results in the final version of the paper.

---

> > ### Comment · Reviewer_8sZj · 2024-11-24
> >
> > 7.
> >
> >    - As suggested you can consider changing the title to be more explicitly focused on "alignment" instead of early fusion.
> >
> >    - As you mentioned this paper is an extension of ClearCLIP. From the comparison between EF-VLA (xattn) and EF-VLA, the most important contribution seems to be ClearCLIP.

---

> ### Author Response · Authors · 2024-11-25
>
> #### 1. Replied in the general response.
>
> #### 3. cluttered/complex/natural backgrounds
> As mentioned in the paper, all our real-world experiments contain 2 or 4 distractors. As shown in Figure 6 (the above two rows), there are multiple objects in the scene. The two rows below just provide illustrative examples.
>
> #### 4. "..., How are N and k chosen? How do they affect the capability of following language instructions, vision-language patch alignment, and generalizing to unseen tasks? Is the concatenation the only way to do this fusion?"
> We already added those details to the revised paper. We didn't explicitly perform parameter search on N and k, but naturally make $N*k$ to be the hidden dimension for the policy transformer, which is set to 512 (768 for our larger model). The vision-language patch alignment is done in the CLIP model, which is independent of N and k. The other capabilities also come from the CLIP alignment, as our xattn and late fusion baselines also adopt the same N and k but fail to generalize. As long as N and k are expressive enough to carry important information (in our experiments, one token per image with N*k=512 ), it should perform reasonably well. For sure, concatenation is not the only way, it's one straightforward implementation of other equivalent choices.
>
> #### 5. Since the paper aims at generalization capability, why didn't directly evaluate it in zero-shot settings?
> First, We don't have access to the pre-training physical scenes, e.g. Google RT Dataset or Bridge Dataset. The pre-trained model (on OXE) cannot zero-shot generalize to our physical setup, which is totally unseen during pre-training. OpenVLA and Octo failed to perform well on this setup after finetuning, not to mention their zero-shot performance.
> Second, we **already evaluate in zero-shot settings** on our own physical setup, with unseen objects, distractors, and language instructions that are unseen in our training dataset.
>
> #### 5+6. "...How can we know your implementation is correct?", "How can I know your long-horizon extension is correct?"
>
> To address your concerns about the baselines, we extend them by changing some modules or hyper-parameters in the config for OpenVLA and Octo, following their instructions in their official codebase (e.g. Octo finetuning with new action horizon at https://github.com/octo-models/octo/blob/main/examples/02_finetune_new_observation_action.py). In Octo's own paper, they didn't observe benefits after increasing context length (Appendix E, "We did not observe benefits of increasing the history length further on the few tasks we evaluated on, though other tasks may benefit."), which is consistent with our observations.
>
> At this moment, we are unable to figure out how to address your concerns about **whether our implementation is correct**. We believe these concerns stem more from subjective opinions rather than an objective evaluation, which we feel goes beyond the bounds of the ICLR code of ethics (https://iclr.cc/public/CodeOfEthics). We respectfully request the reviewer to assess our contribution based on scientific evidence instead of personal opinions. Any deviation from this standard would result in an unfair review.
>
> #### 6. What's the result of the original Octo without extension? Need a case that matches OpenVLA.
>
> The paper already provides the results for the original Octo. We extend that upon your request.
> OpenVLA is a 7B model that requires a long time to train. However, we also don't think it's necessary to match OpenVLA as we already demonstrated the scalability of our method by introducing a larger model that matches Octo, with consistent conclusions. In fact, when OpenVLA compared with Octo, it also didn't match Octo. It should be considered as an advantage that our model has fewer parameters than OpenVLA but can still perform and generalize better.
>
>
> #### 7. "you can consider changing the title to be more explicitly focused on "alignment"... From the comparison between EF-VLA (xattn) and EF-VLA, the most important contribution seems to be ClearCLIP."
>
> Thank you, we will change the title accordingly.
>
> ClearCLIP is an important tool for extracting fine-grained vision-language alignment information at the patch level. Our contribution is to pick the correct method to obtain aligned vision-language representations, and effectively leverage them to train a better robotic generalist policy. For example, SigCLIP and DINO play important roles in OpenVLA, but they are not the contributions of OpenVLA.

---

> > ### Comment · Reviewer_8sZj · 2024-11-27
> >
> > Thanks for your clarifications.
> >
> > 3. The point is about the black background in all examples in Figure 6, not objects. Some visualizations using non-black backgrounds would better highlight the quality of the attention map.
> >
> > 4. Thank you for the explanation of how these values are chosen in this response. It looks like there is only a brief introduction on N and k at L227-229. There seems no explanation in the revised paper and the response. The study on how to choose N and k is still missing. In addition, after carefully checking the left illustration of Figure 2, I think "Vision-Lang Fusion" block should also be labeled as "learnable", because there are learnable parameters when fusing the tokens.
> >
> > 5. Thank you for referring to Octo's source code. It addressed my concern about implementation correctness. However, the architectural difference is there. The difference between EFVLA and naively expanding Octo/OpenVLA is that EFVLA has a token reduction operation. I think this key difference should be highlighted in the paper, and should be clearly studied. This issue also connects to my comment about the technical contribution of the overall paper, which will be discussed in point 7.
> >
> > 6. To scientifically evaluate the result, I'm afraid that I can not make projections without a data point. There are only two data points on scaling so far. (Figure 5 is about scaling the frozen CLIP part). The largest model matches Octo, but has much fewer parameters than OpenVLA.
> >
> > 7. Contribution of the paper
> > The paper was initially posed as "early-fusion" helps "VLA model" "generalize". From my understanding, these are the three main points the paper should focus on. The revision clarified "early-fusion" by "alignment".
> >  - The first part, "alignment". This proposed technique heavily relies on ClearCLIP.
> >  - However, in the second part, "VLA model", I do find the unique things added by the authors, which are reducing vision-language tokens and the design of input tokens. Unfortunately, the token reduction is not well presented and studied. The reduction also does not match the claim of "fine-grained vision-language tokens" in the abstract.
> >   - The third part, due to the difficulty of making experiment settings close (data, architecture, and model size), it's also hard to know whether the generalizability is from fine-tuning a smaller model on a smaller dataset.
> >
> > Overall, the paper is organized to present a novel architecture of VLA model using vision-language alignment, but the key change of representations is purely from ClearCLIP, the unique architectural design is not well studied, and the experimental setting is a bit unfair.
> >
> > However, it is possible to rework the paper to make the presentation and the contributions clear. On top of the "alignment" which we may have agreed with each other, my remaining suggestions are the following:
> >   - Organize the paper in a straightforward way that: 1. directly introduce how ClearCLIP gives you better vision-language alignment as a preliminary section. 2. (your contribution) Introduce how you take advantage of the better alignment from ClearCLIP, highlighting your token reduction and other input designs. 3. Verify 1 and 2 (your contribution) with experiments.
> >
> >   - The title or the main idea of the paper should change from that the vision-language "alignment" helps VLA model generalize to that you propose a new architecture, if you can not show it is a general idea works for some/most of the existing VLA models. If you want to successfully show it is a general idea, apple-to-apple comparisons are the best. I understand it is difficult to obtain their original experimental setups, but please also understand we could not imagine a non-existing result and we should scientifically evaluate the contribution.
> >
> >  - Give up the claim on fine-grained representations. The reduction (N=4) radio is very high. I may understand the fine-grained representation/alignment helps you to select tokens, but it happens in ClearCLIP and you are discarding the majority of the tokens.
> >
> > In conclusion, the paper still requires a major revision, and the technical contribution is minor.

---

> ### Author Response · Authors · 2024-11-28
> **Response**
>
> 3. black background
> We have included attention map examples from the OXE dataset in the Appendix Figure 7 where we show how our method works across different setups and backgrounds.
>
> 4. N and k are hyperparameters that were chosen empirically. We did not extensively tune them, but adopted a simple choice (with N=4 and k=64) to match the hidden dimension of the transformer, which already yielded strong performance. Note that training a neural network requires dozens of hyper-parameters (or even more), and people only conduct ablations on the **key design choices**, otherwise the experiment scale is infeasible. Given that N and k are reasonable values (not abnormally small or large), we believe there are not sufficient reasons to conduct this ablation or reject this work because of that, as these hyper-parameters are **not the key design choices**, not specifically tuned but simply chosen, and not considered as any parts of our contribution.
>
>     The attention pooling layer for compression in the "Vision-Languague Fusion" block is already stated as "learnable" in Line 237 of the latest draft.
>
> 5. We emphasize that EF-VLA can perform token reduction effectively due to the aligned vision-language features. Otherwise, it would be very challenging for the VLA to learn the compression operation from scratch, given the results of Octo, OpenVLA, our LF-VLA, and EF-VLA (xattn) baselines. Therefore, we argue that the major contribution of EF-VLA is the novel architecture design, which uses the pre-aligned vision language features to extract the most important patch tokens, and achieves much higher data efficiency on top of that. While obtaining pre-aligned vision-language features has been studied in ClearCLIP for image segmentation, EF-VLA shows a way of adapting it with the effective token reduction in VLAs for robot learning, achieving much better generalization. We want to emphasize that our way of using vision language features from a pre-trained VLM differs from previous work such as RT-1, RT-2, and OpenVLA as discussed in the related work, and this novelty should be acknowledged.
>
> 6. We argue that OpenVLA is finedtuned on prismatic, which is pre-trained on large VLM datasets. It's unfair for EF-VLA to match the size of OpenVLA, as EF-VLA is only trained on robotics datasets without access to a large VLM dataset. Octo is also trained only on robotics datasets. Since EFVLA and Octo are both trained on the same amount of robotics data (OXE), we argue matching EF-VLA with the size of Octo is a fair comparison. We re-emphasize that OpenVLA doesn't scale up Octo for comparison as well. The latest $\pi_0$ model from physical intelligence (https://www.physicalintelligence.company/download/pi0.pdf) also uses a small action model with a backbone VLM for tokenization, without scaling itself to compare with OpenVLA. The community's objective is to develop a scalable generalist robot policy that is appropriately sized for the task, though not necessarily as large as OpenVLA or other large models.
>
> 7. We changed the paper title to: "EF-VLA: vision language action models with aligned vision language features for better generalization" for clarification.
>
>
> ***The first part, "alignment". This proposed technique heavily relies on ClearCLIP.***
> As in response to 5, we acknowledge the alignment relies on ClearCLIP but we argue EFVLA shows a novel way of using pre-aligned vision language features in VLA models, that achieves better generalization performance.
>
> ***Unfortunately, the token reduction is not well presented and studied. The reduction also does not match the claim of "fine-grained vision-language tokens" in the abstract.***
> See response to 4. By "fine-grained", we meant that the "vision language tokens" can be used for accurate patch-level localization (e.g. finding object-related patches), similar to the definition of "dense" in ClearCLIP. Furthermore, **we believe the reduction operation matches the usage of ClearCLIP very well**. ClearCLIP can give us a very clear correspondence map, which can be effectively compressed into a few important tokens, as suggested by its name ClearCLIP.

---

> ### Author Response · Authors · 2024-11-28
> **Response (cont.)**
>
> ***The third part, due to the difficulty of making experiment settings close (data, architecture, and model size), it's also hard to know whether the generalizability is from fine-tuning a smaller model on a smaller dataset.***
> All three models, EF-VLA, Octo and OpenVLA are pre-trained and fine-tuned on the same data. The architecture design of EF-VLA is our major contribution and we do not see the motivation of changing baselines to use our architecture design. As in response to 6, matching model size to OpenVLA is unfair to EF-VLA. Since EF-VLA (large) has the same model size as Octo, we argue the generalizability is not due to the model size. We additionally argue that showing superior performance over 4 different primitives using a relatively small dataset should be an **advantage**, highlighting the data efficiency of EF-VLA. As we already show the data scaling capability of EF-VLA, we don't see the reason for us scaling data just to make baselines better. We sincerely hope the reviewer can acknowledge that we have made the comparison to the baselines as fair as it can be to the best of our capabilities, and the experiment results in the paper are sufficient to show the superior performance of EF-VLA to the other baselines.
>
>
> Thank you for your comments on editing the paper. Section 3.1 already introduced how ClearCLIP gives better vision-language alignment. As in response to 5, the reason that token reduction can perform well is pre-aligned vision-language features, and our baselines LF-VLA and EF-VLA(xattn) already demonstrated this. We have verified the significance of our model design through extensive experiments to the best of our capabilities.
>
> We have changed the title to reflect the reposition. We hope the reviewer understands it's impossible for everyone to evaluate their models using the exact setups as baselines in the field of robot learning. For example, papers [1, 2, 3, 4, 5 ,6 and more] are all using different setups and non-comparable model sizes. Asking for such experiments is unfair and largely discourages researchers with limited resources. We sincerely ask the reviewer to acknowledge the extensiveness of our experiments and that the significant performance gap between our methods and the baselines is sufficient to show the advantages of the proposed method.
>
> [1] Brohan et al., RT-1: Robotics Transformer for Real-World Control at Scale
>
> [2] Brohan et al., RT-2: Vision-Language-Action Models
> Transfer Web Knowledge to Robotic Control
>
> [3] Kim et al., OpenVLA: An Open-Source Vision-Language-Action Model
>
> [4] Octo Model Team et al., Octo: An Open-Source Generalist Robot Policy
>
> [5] Black et al., $\pi_0$: A Vision-Language-Action Flow Model for General Robot Control
>
> [6] Shridhar et al., Perceiver-Actor: A Multi-Task Transformer for Robotic Manipulation

---

### Official Review · Reviewer_rLZD · 2024-11-01

**Soundness:** 2
**Presentation:** 2
**Contribution:** 2
**Rating:** 3
**Confidence:** 4

**Summary:**

This paper introduces a novel approach, Early Fusion VLA (EF-VLA), which leverages the CLIP model’s capabilities in image-text alignment through early fusion of vision-language (VL) features. EF-VLA differentiates itself by extracting fine-grained VL features early in the process, prior to inputting them into a transformer-based policy network. These fused features are concatenated with attention-pooled text features and embodiment features, forming a comprehensive multi-modal state representation. This representation is subsequently fed into a causal transformer policy for fine-grained action generation. The authors evaluate EF-VLA's performance on a diverse set of tasks within both the LIBERO simulation environment and real-world scenarios. Results demonstrate EF-VLA’s superior performance over several baseline approaches.

**Strengths:**

1. The paper is well-structured and easy to follow.

2. The experimental setup is thorough and convincingly demonstrates the effectiveness of EF-VLA. The inclusion of real-world robotic experiments, alongside evaluations on the LIBERO platform, strengthens the validity of the results and highlights the method's practical applicability.

**Weaknesses:**

1. **Potentially Misleading Title and Incomplete Exploration of Early Fusion**: While the title suggests an exploration of “early fusion,” the fusion approach presented here uses attention at the final layer of the vision encoder before passing features to the downstream transformer-based policy. This choice, although earlier than fusing everything solely within the transformer-based policy, falls short of a deeper fusion study that examines the benefits and trade-offs of fusing features at even shallower layers of the vision encoder. Additionally, while the proposed early fusion method generalizes well to LIBERO and real-world tasks, a broader exploration of early fusion designs across diverse model settings, such as OpenVLA, would strengthen claims about generalizability.

2. **Missing Ablations on Key Design Choices**: The study lacks critical ablations that could clarify the choices made in EF-VLA’s architecture. For instance, although the authors introduce a learnable pooled text embedding \(f'_l\) in the state representation, there is no equivalent learnable visual embedding (\(f'_v\)) included, and the rationale for this asymmetry remains unexplored. Additionally, in the LF-VLA configuration, it is unclear why visual and textual tokens are processed through an attention pooling layer to produce final tokens, rather than directly using classification tokens from each encoder, which would be a more straightforward approach. These omissions leave gaps in understanding the impact of design decisions on model performance.

Due to these concerns, I would not recommend acceptance at this stage. Further work on both the scope of early fusion studies and additional ablations would significantly strengthen the paper.

**Questions:**

Listed in the weakness section.

---

> ### Author Response · Authors · 2024-11-21
> **Response to reviewer rLZD**
>
> We appreciate the reviewer for taking the time to review our paper and give constructive comments. We emphasize the contribution of our work and clarify some commonly raised questions in our general response. Here we address your individual questions as follows.
>
> ##### 1. “Potentially Misleading Title and Incomplete Exploration of Early Fusion”
> In this paper, "early fusion" specifically refers to the alignment of language and vision occurring *before* the policy transformer in a *relatively earlier* stage, whereas "late fusion" is happening in the policy transformer. We revised the paper to make it clear. We adopt ClearCLIP in our approach, but our methods can also benefit from other strategies that fuse features at even shallower layers if they are better than ClearCLIP. For applying early fusion to other models e.g. OpenVLA, although we can modify the SIGLIP encoder in a similar manner, re-training the 7B OpenVLA models within the rebuttal period is challenging due to limited computation resources and time.
>
> We emphasize that our contribution is **not exploring different early fusion strategies, but rather highlighting the fact that early fused vision-language representations can significantly improve the generalization ability of VLA models.** We updated the title to avoid confusion.
>
> ##### 2. “Missing Ablations on Key Design Choices”
> * The purpose of keeping $f\'\_l$ is because the language can specify action primitives (e.g. pick, place, poke and pour), while $f\'\_{vl}$ and $f\'\_{v}$ are both visual representations, which encode important task-relevant information such as object locations. So we need both $f\'\_l$ and $f\'\_{vl}$ (or $f\'\_{v}$) to interact with correct objects with specified primitives. The architecture can be symmetric with $f\'\_l$ and $f\'\_v$ (our LF-VLA baseline). We demonstrate that language-filtered visual representation $f\'\_{vl}$ is better than $f\'\_{v}$ by comparing EF-VLA with LF-VLA both qualitatively (Figure 6) and quantitatively (Table 1, 2). For EF-VLA, we didn't add $f\'\_{v}$ alongside $f\'\_l$ and $f\'\_{vl}$, as $f\'\_l$ and $f\'\_{vl}$ already contain sufficient information to specify and execute a task.
>
> * Late fusion using classification token: We replace the cross-attention on patch tokens with the CLS token, and report the success rates across 7 held-out real-world manipulation tasks in the following table. LF-VLA with CLS token almost failed all the held-out tasks, even worse than LF-VLA (patch tokens). This may be because the CLS tokens on the image don't carry enough detailed visual information for manipulation.
>     | Task | EF-VLA | LF-VLA (CLS) |
>     |-|-|-|
>     | yellowcube-in-black-bowl| 0.6 | 0.1 |
>     | yellow-cube-in-grey-bowl| 0.7 | 0 |
>     | blue-bear-in-pink-bowl| 0.6 | 0 |
>     | reddish-in-grey-bowl| 0.55 | 0 |
>     | black-dog-in-pink-bowl| 0.8 | 0.05 |
>     | blue-sponge-in-pot| 0.4 | 0 |
>     | apple-in-black-bowl| 0.7 | 0 |

---

> ### Comment · Reviewer_rLZD · 2024-11-24
>
> I appreciate the authors’ effort in revising the title to better reflect the problem the paper addresses and for providing additional observations. The new title is an improvement over the previous version. However, I still have significant concerns that I hope the authors can address to strengthen the paper, and here are some of my suggestions:
>
> 1. The authors stated that the contribution is `not exploring different early fusion strategies, but rather highlighting the fact that early fused vision-language representations can significantly improve the generalization ability of VLA models.` To support this claim, I suggest the following improvements:
>
> (a) Apply Early Fusion Strategies Across Multiple Models: Testing early fusion strategies on at least two VLA models (e.g., OpenVLA) would provide stronger evidence of the claim’s general applicability. I understand that retraining models like OpenVLA may be challenging due to time constraints, and I am not requesting retraining of any specific model. However, applying the proposed fusion strategies to any additional VLA models would significantly strengthen the claims, as no such validation is evident in the current version.
>
> (b) Conduct Detailed Ablation Studies on Early Fusion Designs:
> It is crucial to evaluate the current design choices comprehensively. For instance, as mentioned in my original review, 'a deeper fusion study that examines the benefits and trade-offs of fusing features at even shallower layers of the vision encoder.' would be valuable. Evidence from works like LLaVA [1] suggests that the final layer of the CLIP vision encoder might not always be the optimal choice; alternatives like the second last layer could yield better results. Such studies would help identify the best practices for applying early fusion strategies.
>
> (c) Evaluate on Other CLIP-like Models:
> It would be helpful to assess whether similar properties exist in other CLIP-like models, such as SigLIP [2]. This would provide further validation of the paper’s claims.
>
> Without such exploration, the work risks being perceived as a case study rather than a substantial research contribution, limiting its broader impact on the community.
>
> 2. Alternative Paper Formulation:
> Another possible approach would be to benchmark the proposed method against existing generalist VLA models and demonstrate its superior performance under fair comparisons. However, the current paper uses fewer benchmarks than those used in Octo or OpenVLA, making it difficult to justify the general advantage of this design. Pursuing this direction could provide an alternative way to structure the paper and present its contributions.
>
> In conclusion, while the revised title is an improvement, the current version of the paper does not sufficiently support its claims. I recommend further refinement to strengthen its contributions and maximize its benefit to the community. As it stands, I don't think the current version of the paper is ready for the community and I cannot recommend the paper for acceptance.
>
> **Reference**
>
> [1] Liu, Haotian, et al. "Visual instruction tuning." Advances in neural information processing systems 36 (2024).
>
> [2] Zhai, Xiaohua, et al. "Sigmoid loss for language image pre-training." Proceedings of the IEEE/CVF International Conference on Computer Vision. 2023.

---

> ### Author Response · Authors · 2024-11-25
>
> We appreciate your valuable feedback to help us improve the paper. We agree that our method would be more impactful by incorporating studies across a wider range of models and early-fusion strategies. Given the current status of the paper, we would like to follow your suggestions to reposition our contribution as advancing VLA models rather than sorely emphasizing early fusion. We can change the title to "EF-VLA: vision language action models with aligned vision language features for better generalization" to reflect this reposition.
>
> We also acknowledge your concerns regarding the smaller number of benchmarks used in our methods compared to Octo and OpenVLA. Unfortunately, we do not have access to their zero-shot evaluation setups (e.g., platforms used for the Google RT Dataset and Bridge Dataset). To address this, we conducted both in-distribution and zero-shot evaluations, with five primitives and diverse objects in our own scenarios, which involve a comparable number of tasks. We are open to incorporating more physical setups in future work.

---

### Official Review · Reviewer_46Xr · 2024-11-04

**Soundness:** 3
**Presentation:** 3
**Contribution:** 3
**Rating:** 6
**Confidence:** 4

**Summary:**

This paper argues against fine-tuning the vision encoder in manipulation policy learning, proposing instead a straightforward yet effective feature fusion method to leverage features extracted by frozen CLIP. This approach preserves the rich alignment capabilities of Vision-Language Models (VLMs) and enables the model to generalize effectively to new environments without extensive vision encoder fine-tuning.

**Strengths:**

- Clear Motivation and Novel Insight: The paper is well-motivated and clearly written, with a compelling argument for early fusion to decouple the vision-language alignment learning from the manipulation control learning. This insight into separating the two learning tasks highlights a thoughtful approach to using pre-trained VLMs in robotics.
- Simplicity and Effectiveness: A simple fusion method is proposed and the experiments show that it is effective in improving the performance of manipulation policy learning.

**Weaknesses:**

The depiction in Figure 3 is somewhat confusing. The figure seems intended to represent the calculation of image-attended text features through an analogy to the attention mechanism using the CLIP score. However, with attention equations presented on the same page, it is unclear if the query, key, and value refer to the same elements discussed in the figure. A clearer distinction or separate notation could help alleviate this confusion.

**Questions:**

- Choice of CLIP Over Other VLMs: Although the paper mentions other Vision-Language Pretrained models like BLIP in lines 122 - 127, it only discusses why fine-tuning CLIP and modifying CLIP is not desirable and does not provide a detailed rationale for not using other VLPs. Beyond CLIP, models such as ALIGN, BEiT, and CoCa offer potentially richer language capabilities. Could the authors elaborate on why they chose to modify CLIP over these alternatives and discuss whether these models might offer additional benefits in manipulation policy learning?
- What is the learnable "cross-attention pooling" in line 225?

---

> ### Author Response · Authors · 2024-11-21
> **Response to reviewer 46Xr**
>
> We appreciate the reviewer for the time to review our paper and give constructive comments. We emphasize the contribution of our work and clarify some commonly raised questions in our general response. Here we address your individual questions as follows.
>
> ##### 1. “Confusing Figure 3”
>
> Thank you for the suggestions to improve the clarity of Figure 3. Here the key, query, value should be $\hat{f}_l$, $\hat{f}_v$, and $\hat{f}_v$, respectively. They are fused through a CLIP-like similarity score, but not standard attention mechanism. We changed the figure accordingly.
>
> ##### 2. "Choice of CLIP Over Other VLMs, e.g. ALIGN, BEiT, CoCa"
>
> We selected CLIP for its wide usage and community support. In contrast, ALIGN is not open-sourced, BEiT is limited to image-only functionality and cannot provide vision-language representations, and CoCa relies on OpenCLIP, which is trained on a smaller dataset compared to CLIP and offers less flexibility (e.g. for JAX/FLAX users).
>
> Moreover, CLIP has many follow-up works like ClearCLIP, which we adopted as an effective strategy to extract detailed vision-language representations. This approach may not be generally applicable, as the properties of other VLMs remain largely underexplored.
>
> As discussed in the general response, our contribution **is not exporing different early fusion strategies, but rather highlighting the fact that early fused vision-language representations can significantly improve the generalization ability of VLA models.**
>
> ##### 3. "What is the learnable "cross-attention pooling" in line 225?"
>
> To reduce the number of tokens, we use $N$ learnable queries $q$, and keys $k$ and values $v$ from $f_{vl}$, and compute the output using cross attention $X_{attn}(q, k, v)$. We concatenate the $N$ output tokens to one single token, which is $f'_{vl}$. We revised the paper (L226) to make it clear.

---

> > ### Comment · Reviewer_46Xr · 2024-12-03
> > **Respond to Author's Rebuttal**
> >
> > I appreciate the extra results and explanations provided by the authors in the rebuttal. After reviewing the comments from other reviewers and the authors' responses, I would like to provide additional comments and suggestions:
> >
> > ---
> >
> > **1. Title and Interpretation of "Early Fusion"**
> > In the original submission, the concept of "early fusion" led to varying interpretations among reviewers. However, the central idea of the paper is clear: fusing vision and language features before feeding them into the manipulation policy learning module. This allows policy learning to be decoupled from vision encoder fine-tuning. The revised title highlights this central idea effectively, but the term "early fusion" could be avoided entirely in the paper. Instead, referring to the proposed method as "fuse before acting" or similar terminology would be clearer and more consistent.
> >
> > **2. Design Choices for Fusion Method**
> > One major weakness is the lack of experimental justification for the fusion method's design choices. While ablation studies demonstrate optimal performance for the current configuration, they do not provide enough insight into *why* the proposed method is effective or how it might inspire further exploration of fusion methods. This concern was also raised by other reviewers, such as Reviewer 8sZj, who questioned "why single token." While the authors justified this by emphasizing VL correspondence at the patch level, more experiments are needed to validate the structural design (e.g., testing a simple fusion layer or exploring other fusion methods, as suggested by Reviewer rLZD), as well as to justify hyperparameter choices.
> >
> > **3. Novelty Concerns Related to ClearCLIP**
> > Reviewer 8sZj raised concerns about the lack of novelty due to ClearCLIP, but I do not see this as a major issue. ClearCLIP demonstrates the potential of using the attention output from the last self-attention layer in CLIP, and this paper provides a good example of utilizing that output for Vision-Language Action (VLA). The vision behind the paper is correct: leveraging simple and effective feature extraction methods from CLIP-related research with minimal modification and demonstrating their effectiveness in a new task.
> >
> > **4. Broader Applicability Beyond CLIP**
> > All reviewers, including myself, expressed concerns about the method's scope being limited to CLIP. The proposed method, based on ClearCLIP and tailored for CLIP, may not generalize to other models like SigLIP. However, I still find this work valuable, as the "fuse before acting" approach is a significant idea worth exploring in VLA. The authors should provide deeper insights into why and how vision and language features should be fused before being fed into manipulation policy learning. Broadening the scope or generalizability of the method would greatly enhance its impact.
> >
> > **5. Vision-Language Problem Learning vs. VLA Research**
> > My initial question in the original review aimed to address the significant mismatch between vision-language learning and VLA research. CLIP represents one stream of Vision-Language Models (VLMs) that utilizes contrastive VL pretraining, minimizing vision-language fusion to a simple form. However, most vision-language learning approaches include generative VL pretraining to develop strong language capabilities and vision-language alignment. I have highlighted several notable works below, which were either omitted or misspelled in my original review, all of which feature well-documented and openly accessible codebases:
> >
> > - **ALBEF:**
> >   Li, Junnan, et al. "Align before fuse: Vision and language representation learning with momentum distillation." *Advances in Neural Information Processing Systems* 34 (2021): 9694-9705.
> >   *(In my original review, I mistakenly referred to this model as "ALIGN" instead of ALBEF.)*
> >
> > - **BEiT v3:**
> >   Wang, Wenhui, et al. "Image as a foreign language: Beit pretraining for all vision and vision-language tasks." *arXiv preprint arXiv:2208.10442* (2022).
> >   *(In my original review, I intended to refer to BEiT v3, the latest version of BEiT, a strong vision-language model applicable for both single-modality and multimodal tasks.)*
> >
> > - **BLIP-2:**
> >   Li, Junnan, et al. "Blip-2: Bootstrapping language-image pre-training with frozen image encoders and large language models." *International Conference on Machine Learning, PMLR* (2023).
> >
> > ---
> >
> > **Final Judgment and Score**
> > I stand by my recommendation for acceptance of this paper. However, due to the significant areas for improvement outlined above, I have decided to downgrade my score to 6. The paper has great potential, but addressing these concerns—especially around fusion design justification and broader applicability—would make it a stronger contribution to the field.

---

> > > ### Author Response · Authors · 2024-12-04
> > >
> > > Thank you for your detailed comments. We appreciate your acknowledgment to our novelty of "leveraging simple and effective feature extraction" for VLA models.
> > >
> > > The major concern appears to be the design choices for fusion method and broader applicability beyond CLIP.
> > >
> > > As for the design choice, we have added ablation studies on the fusion method including using the cross-attention between the text tokens and the vision tokens and using vision cls tokens as in Appendix. We show both ablations perform worse than the proposed method. In addition, as suggested by Reviewer rLZD, we position this paper to be a VLA model as all outperforming existing VLA models and we re-emphasize that ***our contribution is not exploring different early fusion strategies, but rather highlighting the fact that early fused vision-language representations can significantly improve the generalization ability of VLA models.***
> > >
> > > As for the broader applicability beyond CLIP, we acknowledge the current method relies on the ClearCLIP to provided aligned vision text features. We believe the as the VLM field progresses, better aligned vision text features from other VLMs can benefit the proposed method. This work already shows using pre-aligned vision language features for the VLA models mitigates the burden of the transformer policy to re-learn the vision and language connection and helps maintain vision language alignment capability obtained from larger VLM datasets. We hope this work can demonstrate the potential of a different way of using pre-trained VLM features for VLA training, particularly it's benefits on zero-shot generalization ability. We hope this can encourage people in the community to focus more on aligned vision-language feature extraction and explores diverse ways of using pre-trained VLM for robotics.
> > >
> > > We understand your change of score after reviewing other reviews. But it seems you acknowledge the novelty and potential of the proposed method. Given our re-position of the paper, new experiments and significant performance improvement over the baselines, we sincerely hope you to re-evaluate your judgment.

---

### Author Response · Authors · 2024-11-21
**General response**

### General response:
We thank all the reviewers for the constructive feedback. We are glad to see positive responses appreciating the novelty and contribution of our work, (e.g. “novel insight”, "simplicity and effectiveness" from reviewer 46Xr; “novel approach”, and “thorough and convincing experimental setup” from reviewer rLZD). In the general response, we will address common questions and will clarify other questions in individual comments. **We uploaded the revised paper with changes highlighted in red**.

### 1. The definition of "early fusion"
Both reviewers 8sZj and rLZD raised concerns about the definition of "early fusion". While we acknowledge reviewer 8sZj's perspective that, given a sufficiently large policy network, fusion at any point before the policy could seem "early", our paper specifically scopes "early fusion" to focus on how to better leverage the alignment already present in pre-trained vision-language models (e.g., CLIP). Our approach uses this alignment to filter task-relevant visual features through language features *before* passing them to the policy network. This contrasts with the more common "late fusion" approach, which directly passes all visual and language features to the policy without such pre-filtering. By targeting this formulation, our work is designed to capitalize on the strengths of pre-trained models while addressing the inefficiencies of late fusion strategies.

### 2. Why early fusion helps
EF-VLA utilizes the vision-language alignment capabilities of pre-trained CLIP models to effectively extract task-relevant visual representations. This approach is essential for improving the data efficiency of current VLA models, as CLIP models are trained on massive datasets of image-text pairs that inherently provide well-aligned vision-language representations. In contrast, existing robotic datasets are significantly smaller in scale, making it challenging to achieve comparable capabilities.

### 3. Restricted to CLIP but not general VLMs
As mentioned above, we adopt CLIP as our encoder, because of the wide usage and study of its vision-language alignment capabilities. Other VLMs with similar pre-training objectives can also be applicable, e.g. CoCa as Reviewer 46Xr suggested. Reviewer rLZD also suggested a broader exploration of early fusion designs across diverse models, such as OpenVLA. While it is possible to modify the SIGLIP encoder in a similar manner, pre-training the 7B OpenVLA models within the rebuttal period is challenging due to limited resources and time.

Although we could incorporate more VLMs given sufficient time, we want to emphasize our contribution **is not exploring different early fusion strategies, but rather highlighting the fact that early fused vision-language representations can significantly improve the generalization ability of VLA models.** We also adjust the paper title and contents accordingly to reflect these discussions.

### 4. EF-VLA with more tasks, dataset sizes, and model sizes
* Diverse primitives: We collected new human demonstrations, trained and evaluated models for three new primitives: poking an object, pouring from one cup to a container, and opening or closing drawers. For each primitive, we collect around 200 demonstrations. For more details on the tasks for each primitive, please refer to Appendix Section A in the updated manuscript. We report the performance of all the primitives in Table 3.
* Pre-training on OXE: Both Octo and OpenVLA are pre-trained with OXE dataset. We also pre-train EF-VLA on OXE dataset, and report the updated results in Table 2 and 3.
* Model size scalability: EF-VLA has fewer parameters than Octo and OpenVLA. We enlarge the model size to provide fair comparisons with our baseline models in Table 3.

Through comprehensive experiments, we further demonstrate the consistent advantage of EF-VLA under the above settings. For more details, please refer to our updated manuscripts.

---

> ### Comment · Reviewer_8sZj · 2024-11-24
> **Comment on general response**
>
> Thanks for your clarification and revised paper!
>
> 1. It looks like this work focuses on leveraging alignment in VLM. Thanks for the explanation. I think this term is much clearer than early fusion. I would suggest changing the title and the paper to reflect it.
>
> 2. The arguments are not convincing.
>
>    - Is the value of early fusion only in the low-data regime? Considering the development of the field (scaling of the data and model for example), will early fusion benefit in the long term? Most of your experimental settings are training from scratch or fine-tuning on a small dataset. It's good to show how a large dataset can also benefit from the approach. In Table 3 (revised paper) you have EFVLA trained on the OXE dataset, but it is still fine-tuned on the real-world demonstrations.
>
>    - I would like to discuss whether CLIP creates "well-aligned" vision-language representations. This connects to the third point that how the proposed approach could generalize to different VLMs. Otherwise, the work is studying properties of CLIP which has a limited scope. More VLMs should be tested.
>
>
> 3. See above. I understand the limited time frame but apple-to-apple comparisons are the key to justify the contributions.
>
> 4. I'm confused by the scale of the dataset. Are 200 demonstrations data-efficient for each "primitive"? A missing key baseline is how well a straightforward behavior cloning model (e.g., using DIffusion Policy) can perform under the same amount of training data.

---

> > ### Author Response · Authors · 2024-11-25
> >
> > 1. Thank you for your suggestions. We would definitely consider to emphasize more on vision-language alignment and adjust the paper accordingly.
> > 2. **(a)** First, we disagree that the value of early fusion is only in the low-data regime, and we already showed a large dataset can benefit our model. We already provided the scalability of our model by pre-training on OXE datasets and enlarging the model sizes, both result in better performance.
> >
> >     Second, we also want to emphasize that robotic datasets require substantial human efforts to collect in the physical world. Thus, the scale of robotic datasets cannot match the scale of language or vision datasets in the long run. We think **data efficiency has significant value in robotics**, and our method provides a good solution to this.
> >
> >     Third, even if the scale of the robotic dataset is large (e.g. OXE), people can still have different physical setup which is not covered in the existing dataset, where finetuning is still needed. **Finetuning performance is a necessary and important metric to demonstrate how well the generalist policy is**. If the model cannot adapt to fine-tuning scenarios quickly, there is no point of doing pre-training. Thus, finetuning is a **standard** experimental setup, and both Octo and OpenVLA have done this to demonstrate the model's data efficiency and generalization capabilities in an unseen environment.
> >
> >     Last, the zero-shot experiments for OpenVLA and Octo are performed at the original physical scenes in pretraining OXE dataset (e.g. Google RT Dataset, Bridge Dataset), which are not generally accessible, so we cannot directly test the zero-shot capabilities there. We finetune all models on our own dataset and test the zero-shot capabilities on unseen objects, distractors, and language instructions, which is equivalent to what Octo and OpenVLA did on Google RT Dataset or Bridge Dataset (both in OXE). They cannot generalize on our scenes even after finetuning, not to say zero-shot deployment.
> >
> >     **(b)** This paper aims to show that aligned vision language features improve the generalization ability. However, obtaining such well-aligned vision-language representations requires specific design choices as shown qualitatively in Figure 6. While we believe EF-VLA can benefit from any well-aligned vision language features from other VLAs, obtaining such aligned vision language features from other VLMs remains unexplored by the community and is out of the scope of this paper. We believe in the long run EF-VLA can benefit from the development of the vision language community.
> >
> > 3. Adapting OpenVLA architecture to be early fusion breaks the design initiative of OpenVLA to utilize the pre-trained prismatic VLM. We also disagree that "apple-to-apple comparisons are the key to justify the contributions". We believe comparing the methods as all followed by ablation studies to justify the key design choices are sufficient to demonstrate the contribution.
> >
> > 4. *"Are 200 demonstrations data-efficient for each "primitive"? A missing key baseline is how well a straightforward behavior cloning model (e.g., using DIffusion Policy) can perform under the same amount of training data."*
> > We clarify that 200 demonstrations are per primitive not per task. As shown in the appendix, we have 3 tasks for the pouring primitive and 4 tasks for the poking primitive, resulting in around 50-60 demonstrations per task.
> > It's harder to train multi-task generalist policy than single-task specialists. Our goal is to develop a generalist robotic policy but not a single-task specialist, so we only compared with baselines like OpenVLA or Octo. Note that none of them compared with diffusion policy.
> >
> > In summary, we think **a generalist robotic policy can scale up with dataset, can generalize to unseen tasks in the training physical setup, and can quickly adapt to novel physical setup in a data-efficient manner**. We show our method can satisfy the above. It's not fair for our method or future methods to compare with single-task specialists or evaluate pre-training (e.g. OXE) physical setups that are not accessible.

---

### Meta-Review · Area_Chair_sn7t · 2024-12-23

**Metareview:**

This paper introduces EF-VLA (Early Fusion Vision-Language-Action), a novel model architecture that performs fusion of vision and language features before passing them to the policy network for robotic tasks. The key finding is that by leveraging pre-trained vision-language models' alignment capabilities through early fusion, EF-VLA achieves significantly better generalization performance compared to existing VLA models across both simulated and real-world robotic manipulation tasks. The model keeps the vision-language model frozen while using its aligned representations, helping maintain generalization capabilities without requiring fine-tuning.

However, there are some limitations and concerns about the work. The heavy reliance on CLIP and ClearCLIP raises questions about the broader applicability of the approach to other vision-language models. The token reduction strategy, while effective, appears to contradict claims about maintaining fine-grained representations. Reviewers seemed generally confused about the techical contributions of this work and remain unconvinced after the rebuttal. The discussion suggests divided opinions, with some seeing valuable practical contribution despite technical limitations, while others remained concerned about novelty and experimental comprehensiveness. The AC sees the potential of this work but believe it would be much stronger with a significant revision (possibly a repositioning of this paper + other suggestions).

**Additional Comments On Reviewer Discussion:**

See above.

---

### Decision · Program_Chairs · 2025-01-22

Reject